# Rapid and reliable assessment of methane impacts on climate

Ilissa B. Ocko[1], Vaishali Naik[2] and David Paynter[2]

[1]Environmental Defense Fund, Washington DC, 20009, USA
[2]NOAA Geophysical Fluid Dynamics Laboratory, Princeton, 08540, USA

*Correspondence to*: Ilissa B. Ocko (iocko@edf.org)

**Abstract.** It is clear that the most effective way to limit global temperature rise and associated impacts is to reduce human emissions of greenhouse gases, including methane. However, quantification of the climate benefits of mitigation options are complicated by the contrast in the timescales at which short-lived climate pollutants, such as methane, persist in the atmosphere

as compared to carbon dioxide. Whereas simple metrics fail to capture the differential impacts across all timescales, sophisticated climate models that can address these temporal dynamics are often inaccessible, time-intensive, require special infrastructure, and include high unforced interannual variability that makes it difficult to analyse small changes in forcings. On the other hand, reduced-complexity climate models that use basic knowledge from observations and complex Earth system models offer an ideal compromise in that they provide quick, reliable insights into climate responses, with only limited

computational infrastructure needed. They are particularly useful for simulating the response to forcings of small changes in different climate pollutants, due to the absence of internal variability. In this paper, we build on previous evaluations of the freely-available and easy-to-run reduced-complexity climate model MAGICC by comparing temperature responses to historical methane emissions to those from a more complex coupled global chemistry-climate model, GFDL CM3. While we find that the overall forcings and temperature responses are comparable between the two models, the prominent role of

unforced variability in CM3 demonstrates how sophisticated models are potentially inappropriate tools for small forcing scenarios. On the other hand, we find that MAGICC can easily and rapidly provide robust data on climate responses to changes in methane emissions with clear signals unfettered by variability. We are therefore able to build confidence in using reduced complexity climate models such as MAGICC for purposes of understanding the climate implications of methane mitigation.

## 1 Introduction

Reduced-complexity climate models offer an ideal framework for evaluating greenhouse gas mitigation options if they can accessibly and rapidly reproduce the results of the more complex global chemistry-climate models (CCMs) that include more advanced and comprehensive treatments of chemistry and physics (Meinshausen et al., 2011a). However, there is a critical need to build confidence in the ability of reduced-complexity models to simulate temperature responses to individual greenhouse gases rather than just the suite of climate pollutants, because greenhouse gases have vastly different radiative

properties and atmospheric lifetimes (Myhre et al., 2013; Fiore et al., 2015); it is important to confirm that individual species are represented appropriately if reduced-complexity models are to serve as an effective tool for assessing climate benefits of mitigation actions. This is especially central for the analysis of methane ($CH_4$) mitigation actions, of which the climate policy community has been increasingly focused on (e.g., Shindell et al., 2012; Collins et al., 2018). Therefore, this paper builds on previous evaluations by comparing forcing and temperature responses to historical methane and carbon dioxide ($CO_2$) concentrations from a widely-used reduced-complexity climate model, with that from a state-of-the-art coupled global chemistry-climate model. While it is difficult to compare climate responses of simple models with that of complex ones because of the presence of unforced variability in the latter, we ultimately seek to determine if general temporal patterns and magnitudes are consistent enough to justify the use of a reduced-complexity climate model as a reliable tool for rapid assessment of methane mitigation measures.

Climate change impacts have now been observed on every continent and in every ocean (Stocker et al., 2013). If we want to reduce near- and long-term anthropogenically caused warming, then we need to reduce emissions of several climate pollutants. While limiting long-term climate warming requires drastically reducing $CO_2$ emissions, reducing emissions of short-lived climate pollutants (SLCPs)—specifically, methane and black carbon (BC)—has been identified as one of the most impactful ways to reduce near-term warming (e.g., Ramanathan and Xu, 2010; Shindell et al., 2012; Rogelj et al., 2013; Shoemaker et al., 2013). Methane emissions in particular account for a quarter of the excess energy trapped by human emissions, and today's global anthropogenic methane emissions will have a larger impact on near-term warming than today's global fossil fuel $CO_2$ emissions (based on forcing data provided in Myhre et al., 2013 and references therein; methane emissions provided in EPA, 2012; $CO_2$ emissions provided in IEA, 2015; and radiative efficiency estimates of methane provided in Etminan et al., 2016). Sustained methane emissions will also impact long-term warming (Allen et al., 2016). Furthermore, reducing methane emissions has air quality, health, and food security co-benefits (Shindell et al., 2012; West et al., 2013; Zhang et al., 2016; Melvin et al., 2016).

Most methane mitigation measures are assessed as a comparison to carbon dioxide warming impacts; almost all policy analyses rely on the simple metric Global Warming Potential (GWP) because of its simplicity and ease of use (Fuglestvedt et al., 2010; Ocko et al., 2017). However, GWP is limited in its ability to quantify climate effects because it relies on the integrated impact of a pulse of emissions over a specified time horizon. Because methane and $CO_2$ have vastly different atmospheric lifetimes, their respective climate impacts occur over different timescales. Due to the inherent selection of a single time horizon, GWP is incapable of capturing these important temporal distinctions (e.g., Solomon et al. 2010; Alvarez et al., 2012) unless two time horizons that represent near- and long-term impacts are reported simultaneously (Ocko et al., 2017).

Assessment of SLCP climate impacts over different timescales can be performed using comprehensive global chemistry-climate models (CCMs), however, a full assessment of various SLCP scenarios using sophisticated CCMs is computationally intensive and time-consuming, and forcing perturbations from slight changes in individual species are often too small for the

response signal to be detected among the high unforced internal climate variability present in CCMs (e.g., Ocko et al., 2014). Determining robust climate responses to small-forcing scenarios using CCMs therefore requires a large number of ensemble simulations (Deser et al., 2012). Given that many institutions do not have access to CCMs nor the technical capacity or expertise to run these models, they must rely on partnerships with modelling centres that are often focused on model

development. These characteristics of CCMs reinforces the use of the simple GWP metric for assessments of climate pollutant mitigation measures.

While detailed assessment of regional climate responses can only be provided by complex CCMs, reduced-complexity climate models offer a useful alternative for global changes in major climate characteristics that is far more advanced than GWP but avoids the need for the tremendous amount of computational resources required to perform CCM simulations (and especially

with enough ensemble members to average out unforced variability). These simpler models can rapidly analyse global average climate responses because they are easily accessible and quick to run, thereby providing immediate scientific guidance for mitigation assessments. Further, because they do not include unforced internal variability, they provide clear responses to small forcing scenarios without any noise. .

There are several models that have been developed that fall within this intermediate complexity class—more advanced than

simple metrics but far less sophisticated than CCMs. They range from simplified expressions (e.g., Shine et al. 2005) to more complex chemistry and physics but computations of only a few climate indicators averaged over large spatial domains (e.g., Meinshausen et al. 2011a; Hartin et al. 2014). One of the latter is the freely available Model for the Assessment of Greenhouse-gas Induced Climate Change (MAGICC), initially developed in the late 1980s (Wigley and Raper, 1987, 1992) and routinely updated since (e.g., Meinshausen et al., 2011a). While not meant to replace atmosphere-ocean global climate models

(AOGCMs) and carbon cycle models, MAGICC is a complementary, computationally-inexpensive tool that is capable of efficiently analysing basic climate responses (such as radiative forcing, surface air temperature, and ocean heat uptake) to a suite of emission scenarios. Confidence in MAGICC results comes from a comprehensive effort to match several AOGCMs and carbon cycle models (Meinshausen et al., 2008, 2011a). Evaluations show that MAGICC closely reproduces temperature responses to aggregated forcings from the sophisticated Coupled Model Intercomparison Project CMIP3 atmosphere-ocean

and C4MIP carbon cycle models (Meinshausen et al., 2011c).

While not the only model of its class, the reduced complexity model MAGICC is an especially great resource for mitigation analysis because of its widespread use in international climate reports, and the ability of the user to modify future emissions of every radiatively active species. Therefore, when numerous scenarios exist and need to be evaluated for decision-making, a tool like MAGICC can provide rapid insight into the climate impacts of various options. However, to build confidence in

MAGICC's evaluation of greenhouse gas mitigation strategies, we need to adequately assess its ability to reproduce climate responses to individual greenhouse gases beyond the aggregated forcings. Here, we analyse the capability of MAGICC in simulating climate responses to historical increases (1860-2014) in methane and $CO_2$ by comparing the results with that from a state-of-the-art coupled chemistry-climate model, the National Oceanic and Atmospheric Administration (NOAA)

Geophysical Fluid Dynamics Laboratory (GFDL) CM3 model, which has been shown to adequately reproduce historical temperature trends (Golaz et al. 2013, Griffies et al., 2011; Donner et al., 2011; Winton et al., 2012; John et al., 2012; Levy et al., 2013). While it is difficult to compare simpler models with sophisticated ones for scenarios with small forcings – due to high interannual variability built into the latter – it is nevertheless important to do so because of the more advanced and comprehensive chemistry and physics in the more complex models.

We compare the response of the two models to assess similarities and differences, seeking to determine (i) if the forcings/temperature response is comparable; (ii) if the complexity of the CCM provides any benefits over the simple model; and (iii) whether the lack of variability in the simple model provide any advantages over the CCM when looking at small forcing amounts. Our goal is to build confidence in the simulation of the climate response to methane in order to justify future use of reduced complexity climate models, such as MAGICC, to assess the climate impact of methane emissions mitigation scenarios. In this analysis, we add to previous evaluations by showing a high correlation between CM3's and MAGICC's radiative forcing and surface air temperature responses to changes in either $CO_2$ or methane in isolation, despite large unforced variability in CM3, thereby strengthening confidence in MAGICC's simulation of climate responses to individual greenhouse gases with vastly different radiative properties and lifetimes.

## 2 Models and Simulations

### 2.1 MAGICC model description

We use MAGICC v.6 version developed in 2011 (http://www.magicc.org/download). MAGICC represents the complex coupled carbon-cycle climate system as a hemispherically averaged upwelling-diffusion ocean coupled to an atmosphere layer and a globally averaged carbon cycle model. The atmosphere has four boxes (one over land and one over ocean for each hemisphere) and is coupled to the mixed layer of the ocean hemispheres. The default number of ocean layers in each hemisphere is 50 including the mixed layer (though users can select the number of levels), and heat exchange is driven by vertical diffusion and advection. The terrestrial carbon cycle model is a globally integrated box model with one living plant box and two dead biomass boxes (one for detritus and one for organic matter in soils). The terrestrial carbon cycle does not feedback into carbon dioxide concentrations in the atmosphere. The sea-to-air carbon flux is determined by the partial pressure differential for carbon dioxide between the atmosphere and surface layer of the ocean.

From 1765–2005, the MAGICC v.6 radiative forcing is driven by global-mean concentrations of greenhouse gases (carbon dioxide, methane, nitrous oxide, ozone-depleting substances and their replacements); prescribed regional direct aerosol radiative forcings (sulphate, black and organic carbon, sea salt, mineral dust); land-use, volcanic, and solar radiative forcings; prescribed black carbon on snow radiative forcings; emissions of tropospheric ozone precursors (carbon monoxide, nitrogen oxides, non-methane volatile organic carbon); and indirect (first and second) aerosol forcings calculated from prescribed

regional aerosol optical depths (parameterizations described in detail in Meinshausen et al. (2011a)). For 2006 to 2014, the model is driven by emissions of gases and aerosols taken from the Representative Concentration Pathway (RCP8.5) scenario to capture a business-as-usual trajectory. Climate responses (such as surface air temperature) are provided as global annual averages and also across four spatial boxes (over land and ocean and by hemisphere).

5 Historical greenhouse gas concentrations are from Meinshausen et al. (2011b); prescribed aerosol forcings and land-use historical forcings are from the National Aeronautics and Space Administration (NASA) GISS model (http://data.giss.nasa.gov/); solar irradiance is provided by Lean et al. (2010); and historical emissions of ozone precursors are from Lamarque et al. (2010). Present-day and future (2005–2100) forcings are driven by emissions of gases and aerosols, and are taken from the Representative Concentration Pathway (RCP8.5) scenario to capture a business-as-usual trajectory, though 10 we restrict our analysis here to climate responses from 1860–2014. Carbon dioxide radiative forcings are calculated using a standard simplified expression (Shine et al. 1990 with updated scaling parameter from Myhre et al. 1998). Methane radiative forcings are calculated using a radiative efficiency parameter in conjunction with standard simplified expressions from Myhre et al. (1998), and accounts for overlap between methane and nitrous oxide absorption bands.

For the most recent version of MAGICC, seven key climate parameters were calibrated to match 19 AOGCMs used in the 15 Intergovernmental Panel on Climate Change (IPCC) Fourth Assessment Report AR4 (see Meinshausen et al., 2011a). The parameters include: equilibrium climate sensitivity, land-ocean warming ratio at equilibrium, vertical diffusivity in the ocean, sensitivity of feedback factors to radiative forcing change, sensitivity of vertical diffusivity at mixed layer boundary to global-mean surface temperatures (i.e., thermal stratification), land-ocean heat exchange coefficient, and an amplification factor for the ocean to land heat exchange. The MAGICC parameter set that best reproduces surface air temperatures and heat uptake of 20 each AOGCM is determined via an optimization routine with 1000 iterations to find the combination that minimizes the squared differences between low-pass filtered time series. The effective climate sensitivities in MAGICC v.6 vary over time due to spatially non-homogeneous varying feedbacks, until they reach the equilibrium climate sensitivity. The equilibrium climate sensitivity input into MAGICC depends on which AOGCM calibration is used; they range from 1.9 to 5.73 °C across all 19 models, with a mean (median) of 2.88 °C (2.59 °C). Multi-model-ensembles are generated by running each simulation 25 for all 19 AOGCM calibrations, which we refer to as "physics-driven ensemble members." The user of the downloaded MAGICC model can select which parameters to use for each simulation.

While the MAGICC model is particularly well-calibrated to more sophisticated models, the realism of MAGICC results relies on the realism of GCMs, which have their own sets of limits and uncertainties. Further limitations of MAGICC include incomplete knowledge of forcing patterns, unknown responses outside of the calibrated range, limited set of climate responses 30 evaluated (such as temperature and heat uptake but not precipitation), reliance on a high level of parametrization (such as cloud feedbacks tuned to match those of more sophisticated GCMs), and possible errors in the data used for calibration. In addition, although the model is freely available, it is not open source. However, despite these limitations, MAGICC has been shown to

reasonably reproduce climate responses to all-forcing scenarios (Meinshausen et al., 2011a) and is one of the most prominent reduced complexity climate models in use.

## 2.2 CM3 model description

We employ the GFDL global coupled atmosphere-ocean-chemistry model (GFDL-CM3; Donner et al., 2011; Griffies et al., 2011) to assess the climate response to historical changes in methane and $CO_2$. CM3 uses a finite-volume dynamical core on a cubed-sphere horizontal grid composed of six faces; each face includes $48 \times 48$ grid cells. The size of the grid cells range from 163 km at the corners to 231 km near the face centres. In the vertical, the model domain extends from the surface up to 0.01 hPa (86 km) with 48 vertical hybrid sigma pressure levels. The model simulates tropospheric and stratospheric chemistry interactively over the full vertical domain, with simulated ozone and aerosols influencing radiation calculations (Naik et al., 2013; Austin et al., 2013). Ensemble members for CM3 are generated by employing different sets of stochastically-selected initial conditions (discussed in more detail in Sect. 2.3.), which we refer herein as "initial condition-driven ensemble members." The equilibrium climate sensitivity of CM3 is 4.8 K (Paynter et al., 2018), which is in the range of the MAGICC calibration models but higher than the median and mean.

Global mean concentrations of well-mixed greenhouse gases (WMGHGs), including carbon dioxide, nitrous oxide, methane, and ozone-depleting substances (ODSs) are specified for radiation calculations for the historical period (1860-2005) from Meinshausen et al. (2011b) and for the period 2006 to 2014 following the Representative Concentration Pathway (RCP8.5) scenario. Within the chemistry module, global mean concentrations of methane are prescribed at the surface as the lower boundary condition and are allowed to undergo chemistry everywhere else in the model domain. Radiation calculations do not see the full three-dimensional methane field (simulated in the chemistry module) and only employ the global-mean concentrations, however, changes in ozone and water vapour are seen by the radiation. Further, CM3 $CO_2$ concentrations do not get altered by reactions that occur in the model.

CM3 is forced with emissions of short-lived species including ozone precursors, and aerosols and their precursors, volcanic aerosols, solar radiation, and land-use change as described in detail by Donner et al. (2011) and Naik et al. (2013). Anthropogenic emissions, including from biomass burning and ships, for the time period 1860–2005 are from the dataset of Lamarque et al. (2010) developed in support of the Couple Model Intercomparison Project Phase 5 (CMIP5). For years 2006–2014, anthropogenic emissions follow the RCP8.5 scenario. Natural emissions of all precursor species, except isoprene, are included as described by Naik et al. (2013). Biogenic isoprene emissions are calculated interactively, as described by Lin et al. (2012), based on the Model of Emissions of Gases and Aerosols in Nature (MEGAN) (Guenther et al., 2012). 'Explosive' volcanic eruptions are imposed via a time series of volcanic optical properties rather than from direct injection of sulphur into the stratosphere (Stenchikov et al., 2006; Donner et al., 2011).

Shortwave and longwave radiation algorithms in CM3 are described in Freidenreich and Ramaswamy (1999) and Schwarzkopf and Ramaswamy (1999), respectively, with some modification to enhance computational efficiency (GAMDT 2004). The shortwave algorithm includes 18 bands in the solar spectrum, and the longwave algorithm includes eight bands. Shortwave radiation parameterizations account for absorption by water vapour, carbon dioxide, ozone, molecular oxygen; molecular scattering; and absorption and scattering by aerosols and clouds. The longwave radiation parameterizations account for absorption and emission by water vapour, carbon dioxide, ozone, nitrous oxide, methane, halocarbons (CFC-11, CFC-12, CFC-13 and HCFC-22), aerosols, and clouds. Aerosols included are sulphate, carbonaceous (black and organic carbon), dust, and sea salt.

CM3 includes explicit representation of both the direct and indirect aerosol effects on radiation. For the calculation of the direct effect of aerosols on radiation, physical and optical properties of sulphate, black carbon, organic carbon, sea salt, and dust are considered in the model (Donner et al., 2011). Sulphate and black carbon are assumed to be internally mixed while all other aerosols are assumed to be externally mixed for radiation calculations. To account for the indirect effect of aerosols via aerosol-water cloud interactions, the model treats water soluble aerosols, including sea salt, and organic aerosols as cloud-condensation nuclei (CCN) allowing a physically based parameterization of CCN activation (Ming et al., 2006). The model does not consider the reduction in surface albedo caused by the deposition of black carbon on snow-covered surfaces.

### 2.3 Simulations

Three historical simulations are run for both MAGICC and GFDL CM3 to derive climate responses to isolated $CO_2$ and methane concentrations, respectively. MAGICC was run from 1750 to 2100 by default, and CM3 was run from 1860 to 2014. As shown in Table 1, the direct runs for both models include an all-forcing simulation with all forcings varying with time except land-use; a simulation with $CO_2$ concentrations held at 1860 levels; and a simulation with methane concentrations held at 1860 levels. Subtracting temperature responses of the two latter runs from the former yield $CO_2$-only and methane-only climate responses, respectively (see Eqs. (1) and (2)). The same equations hold for the forcings as well.

$$\Delta T_{CO_2} = T_{AllForc} - T_{CO_2 1860}, \tag{1}$$

$$\Delta T_{CH_4} = T_{AllForc} - T_{CH_4 1860}, \tag{2}$$

For MAGICC, each simulation is run for all 19 AOGCM-calibrated configurations; each 350-year integration took approximately one second to run on a modern PC with a three GHz CPU processing speed. We use default MAGICC gas and aerosol properties, but update tropospheric ozone radiative efficiency and methane atmospheric lifetime to IPCC Fifth Assessment Report (AR5) values (Myhre et al., 2013; Stevenson et al., 2013) to reflect the latest science. (Note that the updated atmospheric lifetime only impacts the model from 2006-2014 as it is driven by emissions and not concentrations during this period.) However, we do not include newer estimates of methane radiative effects that account for shortwave absorption in

addition to longwave absorption (Etminan et al., 2016) to be consistent with the CM3 model that only includes longwave effects. Including the shortwave component increases methane's radiative efficiency by over 20%. Further, we specifically do not tune MAGICC model climate and forcing properties to match that of CM3 because we are assessing how a "standard version" of the reduced-complexity climate model compares with CM3; the goal is not to match MAGICC to CM3 but to

assess whether a downloaded version of MAGICC broadly behaves similarly to CM3. However, two of MAGICC's physics-driven ensemble members are derived from two predecessors of CM3: CM2.0 and CM2.1 (Delworth et al., 2006).

The set-up of GFDL-CM3 simulations conducted here was similar to that adopted for simulations performed in support of the CMIP5, except we obtained initial conditions from a longer preindustrial control (3000 years). Three-member initial condition-driven ensembles of transient CM3 simulations were performed with each ensemble member initialized stochastically at

different points in the preindustrial control simulation. Each 155-year integration of CM3 took about 15 days to complete on the NOAA's Remotely Deployed High Performance Computing System (RDHPCS) machine known as "Gaea" running on 464 processors. While three ensemble members is relatively small, we are limited by computational resources and studies have shown that forced changes in air temperature, as opposed to changes in atmospheric circulation and precipitation, can be detected with fewer ensemble members (Deser et al., 2012).

To compute CM3 radiative forcings for $CO_2$ and methane (direct and indirect) that are closest to the definition used by MAGICC (the forcing at the tropopause after stratospheric temperature adjustment), we performed simulations with the atmosphere-only version of CM3—AM3. The model configuration of AM3 was exactly the same as CM3 except AM3 model integrations over the period 1870 to 2014 were performed with observed sea-surface temperature and sea-ice cover (Rayner et al., 2003), and therefore do not include an ensemble driven by different initial conditions. Through the additional AM3

simulations, we were able to diagnose transient effective radiative forcing (ERF) (the change in net radiation balance at the top-of-atmosphere (TOA) following a perturbation to the climate system taking into account any rapid adjustments (Shine et al., 2003; Myhre et al., 2013) due to $CO_2$ and methane. Transient ERF calculated in this way follow the proposed protocol for the AerChemMIP (Collins et al., 2016). While RF does not capture the full alterations in the energy balance, ERF is more uncertain than RF because it involves multiple climate interactions (Forster et al., 2016). However, several studies have found

that ERF and RF are nearly equal for many situations, and especially for increased concentrations in $CO_2$ and methane (Myhre et al., 2013).

To separate the effect of methane due to its influence on ozone and water vapour (indirect effects) from its effect on radiation (direct effect), we ran two more simulations for MAGICC with methane chemistry turned off (an all-forcing run and methane held at 1860 levels run with methane chemistry turned off for both). Equation (3) shows how the direct methane forcings were

calculated for MAGICC; subtraction between methane-only forcings and the direct forcings yielded the indirect responses to methane. We also ran two more simulations for AM3/CM3 with methane radiation calculations or chemistry held constant beyond 1860, respectively. Equations (4) and (5), respectively, show how the direct and indirect methane forcings were

calculated in AM3. While we only show forcing calculations here via AM3 simulations with fixed sea surface temperatures, we also ran the simulations for the fully coupled CM3 model.

$$\Delta F_{CH_4,direct\,(MAGICC)} \;=\; F_{CH_4 nochem} - F_{CH_4 1860 nochem} \; , \tag{3}$$

$$\Delta F_{CH_4,direct\,(AM3)} \;=\; F_{AllForc} - F_{CH_4 rad 1860} \; , \tag{4}$$

$$\Delta F_{CH_4,indirect\,(AM3)} \;=\; F_{AllForc} - F_{CH_4 chem 1860} \; , \tag{5}$$

The global mean historical concentrations of $CO_2$ and methane used by the models to calculate radiative forcings and therefore temperature changes are shown in Fig. 1. (Note that concentrations are prescribed for MAGICC only through 2005, and then emissions inputs drive the model thereafter; however, the resulting concentrations from these emissions are consistent with that input into CM3.) Results for both models are presented as an average of the individual ensemble members (initial condition-driven ensemble members for CM3 and physics-driven ensemble members for MAGICC). Surface air temperatures are taken to be 2 meters above the surface. For both models, we calculate temperature changes as the difference between temperatures in year $t$ compared to that in 1860.

A key difference between AM3/CM3 and MAGICC is that the full GCM has internally generated unforced variability. This occurs both when the model is coupled (CM3) and run with prescribed sea surface temperatures and sea ice (AM3). The variability can be dampened by applying a smoothing to the annual time-series. However, too long of a smoothing period removes much of the decadal level forcing that we hope to uncover in this study. Therefore, we employ a five-year smoothing average to AM3/CM3 results to filter out some of the internal variability. Additionally, to better quantify and isolate the role of unforced variability in the AM3/CM3, we run control experiments of each with fixed forcing. For CM3 we ran a 500 year control simulation with all radiative forcing held constant at 1860 level. For AM3 we ran a shorter 200 year control run, with all radiative forcing held fixed at 1860 with annually repeating monthly averaged sea surface temperatures and sea ice characteristics taken from 30 years of the CM3 control run.

## 3 Results and Discussion

Here we analyse AM3/CM3's and MAGICC's radiative forcing and surface air temperature responses to changes in either $CO_2$ or methane in isolation.

Given that an important difference between AM3/CM3 and MAGICC results is the role of unforced variability in AM3/CM3, we first analyse the magnitudes of unforced variability in both AM3 and CM3. Although initial condition-driven ensemble member means and/or running averages are employed to dampen out some of the variability in AM3/CM3, it still plays a large

role in forcing and temperature responses. CM3, in particular, has been shown to produce magnitudes of variability on the upper end of CMIP5 models (Brown et al., 2015).

The results of the control simulations with constant preindustrial (1860) external forcings are shown in Fig. 2. In the case of AM3 unforced changes in net radiation at the top-of-atmosphere for all-sky conditions range from -0.18 to 0.21 W m$^{-2}$ with a

standard deviation of 0.07 W m$^{-2}$ for a five-year running mean. We find the maximum swing between two consecutive five-year means to be 0.35 W m$^{-2}$. Sources of unforced variability in AM3 include a mixture of land snow/ice cover variability, clouds, and just year-to-year variability in the meteorology; soil moisture may also play a role. For CM3, unforced internal dynamics yield temperature responses ranging from -0.27 to 0.24 °C for five-year running means with a standard deviation of 0.1 °C. We find the maximum swing between two consecutive five-year means to be 0.2 °C. The variability is driven by

interactions among the ocean-atmosphere-land systems. While unforced variability is a key component to modelling the climate system, it can mask or amplify responses to external forcings over short timescales (e.g., Brown et al., 2017). This makes it difficult to clearly assess responses to small external forcings, and provides further motivation for using simpler models like MAGICC for analysis of small forcing scenarios.

### 3.1 Radiative Forcing

Figure 3 shows the global-mean radiative forcings (RF) in response to the all-forcing scenario as well as forcings attributed to isolated $CO_2$ and methane concentrations, respectively. Note that AM3 forcings are taken as top-of-atmosphere and include rapid adjustments in the troposphere in addition to the stratosphere, and therefore are considered an effective radiative forcing, while MAGICC derived RF is calculated at the tropopause and only considers stratospheric temperature adjustment. While these are the standard forcing calculation methods for both types of models, we emphasize that comparing values of AM3 ERF

to MAGICC RF can only allow for comparisons in broad patterns and relative magnitudes, especially because of large variability in ERF values when averaged over one to five year timescales (Forster et al., 2016) as discussed above.

MAGICC methane and $CO_2$ isolated forcings are much smoother than that of AM3 because of the lack of unforced variability in MAGICC. Some unexpected features in AM3 forcings (such as negative forcings in the earlier years despite increasing atmospheric concentrations) are likely due to unforced variability. Using the MAGICC forcings as a benchmark for a signal

due to forced changes only, we find that nearly all of the deviations of AM3 fall within the range of internal variability as derived from the control simulation: 0.35 W m$^{-2}$. However, despite the slightly different forcing definitions and the unforced variability in AM3, all results are strongly correlated between AM3 and MAGICC (All-forcing r = 0.81, $CO_2$-only r = 0.96, $CH_4$-only r = 0.93).

In the present-day (model year 2014), AM3 and MAGICC yield an all-forcing ERF and RF of 2.0 and 2.5 W m$^{-2}$, respectively;

note that land use is held constant in this analysis. This is consistent with the IPCC (2013) values that show an all-forcing ERF of 2.3 W m$^{-2}$ in 2011 (Myhre et al., 2013). The magnitudes for the AM3 and MAGICC all-forcing radiative forcings are offset

after 1960 (-1 W m$^{-2}$ in 1960). This is due to AM3's strong aerosol indirect forcing (Golaz et al., 2010) beginning around this time when aerosol emissions in the mid-latitudes increased rapidly (Lamarque et al., 2010).

Isolating $CO_2$ and methane's contribution to overall forcings (Fig. 3), MAGICC RF is reasonably consistent with the AM3 ERF evolutions throughout the 20th Century. Preindustrial to present-day forcings for $CO_2$ and methane simulated by AM3 and MAGICC are similar to those given by IPCC (Myhre et al., 2013), (IPCC values: 1.68 from $CO_2$ emissions and 0.97 W m-2 from methane emissions in 2011 relative to 1750 levels), albeit there are important differences, including baseline years (1750 for IPCC and 1870 for AM3 and MAGICC in this study to match that from AM3) and time series of atmospheric concentrations (Myhre et al., 2013). While the same radiation expressions are used for IPCC and MAGICC for $CO_2$ and methane atmospheric concentrations, the representation of tropospheric ozone chemistry and its radiation effects in MAGICC is extremely simplified due to hemispheric averages in a four-box atmosphere. For a short-lived climate forcer that is highly spatially variable, this is a vastly different treatment than that by the IPCC, which employs multi-model assessments for tropospheric ozone forcings. We find that our direct methane forcing in MAGICC in model year 2011 is 0.45 W m$^{-2}$, extremely close to the IPCC's forcing of 0.48 W m$^{-2}$ from changes in methane concentrations alone (recall however different baselines) (Myhre et al., 2013). However, when methane interactions with other chemical species are accounted for, MAGICC estimates a forcing of 0.7 W m$^{-2}$ attributed to changes in methane compared to the IPCC's value of 0.97 W m$^{-2}$ in 2011 (Myhre et al., 2013).

In MAGICC, methane's RF is consistently around half the value of that by $CO_2$. In AM3, methane's ERF is much closer to that of $CO_2$ until the year 2000 and beyond where they diverge. While this divergence is consistent with global atmospheric methane concentrations levelling off for about a decade in the mid-1990s to mid-2000s, before rapidly increasing from 2007 onwards (Fig. 1), further simulations are required (such as more ensemble members or adjustments to input conditions) in order to determine if the close methane and $CO_2$ ERFs before 2000 are an artefact of unforced variability or a substantiated feature. Based on our analysis of unforced variability in AM3, it is quite possible that they are features of internal variability.

Methane's role in radiative forcing can be divided into direct contributions via warming by methane as a greenhouse gas, and indirect contributions via production of other greenhouse gases (mainly tropospheric ozone) as it oxidizes to $CO_2$ in the atmosphere. Figure 4 compares the direct and indirect methane forcings from MAGICC and AM3, calculated via Eqs. (3) and (4), respectively. The results from AM3 further highlight the role of unforced variability in complicating perceived forcings from small concentration changes; the seemingly large swings in AM3 forcings deviate from that of MAGICC by around 0.25 W m$^{-2}$ at most, which is within the realm of unforced variability (see Fig. 2). While correlation coefficients show consistency between MAGICC and AM3 (direct r = 0.87; indirect r = 0.78), the strong variability in AM3 makes comparisons of magnitude difficult. MAGICC attributes around 35% of methane's present-day total radiative forcing to indirect effects, similar to the IPCC's attribution of 34% (Myhre et al., 2013). AM3 shows magnitudes of indirect forcings in the present-day that are around 30-50% of the total methane forcing, depending on the year; this variation is due to unforced variability.

## 3.2 Global Surface Air Temperature Change

To build confidence in the simulation of surface air temperature by both MAGICC and CM3, we compare the model results with 20th Century reconstructions of surface air temperature, of which several datasets are available. Figure 5 shows the historical global-mean surface air temperature responses to changes in all-forcings in MAGICC and CM3 compared with

NOAA and National Aeronautics and Space Administration (NASA) time series of global surface temperature anomalies, freely available online (https://www.ncdc.noaa.gov/cag/time-series/global and https://data.giss.nasa.gov/gistemp/). Following NOAA's methodology (NOAA, 2017), we compute the $20^{th}$ Century average temperatures in MAGICC, CM3, and NASA, and calculate the annual temperature departures from this baseline.

The two observational datasets are perfectly correlated (r = 1.00). MAGICC and CM3 both have high correlations with NOAA

and NASA data, although MAGICC's are higher (MAGICC r = 0.92 (NOAA) and 0.93 (NASA); CM3 r = 0.76 (NOAA) and 0.75 (NASA)). Consistent with Fig. 3, CM3 shows lower temperature responses post-1960 due to the strong effect of aerosols (Golaz et al. 2013). We note, however, that the 'lingering' temperature response in CM3 to major volcanic eruptions is an artefact of the five-year running mean smoothing process; this is why CM3 temperature responses to volcanic eruptions persist longer than what is seen in the observational records and by MAGICC. This is not found, however, to considerably impact the

correlation coefficients between the CM3 data and the NOAA/NASA data. Overall, the general temperature anomaly temporal patterns reveal that both models adequately reproduce surface air temperature, providing confidence in both climate models of differing complexity levels.

The global mean surface air temperature responses attributed to $CO_2$ and methane forcings are shown in Fig. 6, calculated via Eqs. (1) and (2), respectively. The correlations of the ensemble-means (19 physics-driven ensemble members for MAGICC

and three initial condition-driven ensemble members for CM3) are extremely high ($CO_2$ r = 0.98; methane r = 0.92). Figure 6 also shows individual CM3 initial condition-driven ensemble members and the range of MAGICC responses from all 19 AOGCM calibrations; however, we do not include MAGICC's highest climate sensitivity physics-driven ensemble member as the responses were a clear outlier to the rest of the members. Further, recall that the equilibrium climate sensitivity in CM3 is larger than the mean/median in MAGICC, and therefore we expect differences in the ensemble member-averaged responses

from this characteristic alone.

We find that both CM3 and MAGICC attribute a nearly 1 °C rise in temperature from 1860 to 2014 from rising $CO_2$ concentrations (CM3: 0.9 °C; MAGICC: 0.9 °C). For methane, CM3 suggests a rise of 0.5 °C and MAGICC suggests a rise of 0.4 °C, consistent with the larger methane forcing in CM3 (Fig. 3). It is important to note that cooling from aerosols mask some of the warming that we otherwise would be experiencing from $CO_2$ and methane, which is why the combined warming

from $CO_2$ and methane is larger than today's observed warming.

Two major features of the temperature response to methane in CM3, that are not present in MAGICC, further highlight the difficulty of extracting a small signal (and with a small ensemble) given the size of the unforced variability (Fig. 6); methane's

forcing is considerably smaller than that of $CO_2$, making it difficult to extract a temperature response from the variability. The first feature is a global mean cooling response to methane forcings around 1900 to 1915, which is strongly apparent in two of the three initial condition-driven ensemble members. This cooling is likely a lagged response to negative methane ERF (at most -0.15 W m$^{-2}$) from 1895 to 1900, seen both in the direct and indirect methane forcings (Figs. 3 and 4). The second feature

is a strong warming signal in response to methane from 1980 to 1995, followed by cooling through 2000; while this is consistent with AM3 RFs (Fig. 3), the feature is more pronounced in the temperature response. Both of these features fall within the range of annual temperature swings due to unforced variability in CM3 (at most around 0.2 ℃ for a five-year running mean). Therefore, we cannot conclude that they are robust responses to methane, but rather serve as a further example of why CCMs are difficult to employ for small individual forcings and the need for large ensembles.

To dig into these features further, we analyse regional surface air temperature responses to $CO_2$ and methane isolated forcings (Fig. 7). Methane-induced cooling between 1900 and 1915 is strongest in the Southern Hemisphere and especially over Southern Hemisphere oceans. On the other hand, the large methane warming in CM3 around 1990 is most prominent in the Northern Hemisphere, over both land and ocean.

When the global mean responses are parsed out by region (Fig. 7), the highest surface air temperature responses to methane

and $CO_2$ are found over land in the Northern Hemisphere, with temperatures from $CO_2$ rising by well over 1 ℃ from 1860 to 2014 in both models. There is high correlation between MAGICC and CM3 for all regions. We expect and find methane correlations between the two models to be slightly lower than $CO_2$ because methane has more complex chemical interactions in the atmosphere than $CO_2$ that introduce more degrees of freedom than $CO_2$, and are also potentially more simplified in MAGICC. We also find that correlations in the Southern Hemisphere are lower than in the Northern Hemisphere, especially

for methane.

As seen and discussed earlier in Fig. 3 forcings, there are several time periods when the methane temperature responses are comparable in magnitude to that by $CO_2$ in CM3 global mean and regional responses (Fig. 6 and 7). We see this for all initial condition-driven ensemble members, and it is consistent with AM3 RFs (Fig. 3). In the ensemble mean, the comparable warming magnitudes between 1940 and 1950 are consistent with the rate of growth of $CO_2$ concentrations slowing down while

methane concentrations consistently increase (Fig. 1).

Also discussed earlier and in  contrast to the $CO_2$ and methane concentration trends from 1940-1950, the methane concentration growth rate slows down in the 1990s while the $CO_2$ concentrations consistently increase (Fig. 1). This is reflected in the CM3 temperature trends in addition to forcings (Fig. 3) as a divergence in the magnitude of temperature responses between methane and $CO_2$ to where they stand in the present-day, with $CO_2$ yielding twice as much warming in 2014 as methane (Fig. 6).

**4 Conclusions**

The purpose of this study is to enhance confidence in reduced complexity climate models, in the context of simulating temperature responses to methane and $CO_2$ atmospheric concentrations. We use the freely available (but closed source code) and computationally efficient model, MAGICC, for our analysis, motivated by the need to determine a quick and accessible, yet reliable, method for analysing impacts of future changes in methane emissions on climate warming. Given that

sophisticated coupled climate-chemistry models are generally inaccessible, time-intensive, and often employ high internal variability, they can be unsuitable for analysis of methane mitigation strategies when emissions changes are small, when many mitigation scenarios are considered, and when a large extent of parameter sensitivities are to be investigated. Employing a model like MAGICC, rather than resorting to simple GWP metrics, would significantly enhance the accuracy of mitigation assessments while still using basic infrastructure and providing immediate guidance for decision making.

To determine MAGICC's reliability for methane analysis, we performed several sets of experiments using MAGICC and CM3—all forcing with both time-varying natural and anthropogenic forcings but land-use held constant; simulations where $CO_2$ and methane concentrations are held constant at 1860 levels, respectively; and a simulation to isolate methane indirect effects resulting from its influence on ozone and water vapour (for MAGICC, we turned off methane chemistry; for CM3, we held methane radiative effects at 1860 levels). We also ran simulations using the atmosphere-only version of CM3, AM3, to

calculate radiative forcings in response to the four sets of experiments. Finally, we ran control simulations for AM3 and CM3 to determine the role of unforced variability in influencing climate responses.

Both CM3 and MAGICC models adequately reconstruct surface air temperature records from NOAA and NASA from 1860 through 2014, especially for 1950 onwards. For isolated forcings, overall temporal patterns were consistent between MAGICC and CM3 temperature responses to methane and $CO_2$, including for indirect effects via methane chemical reactions. Correlation

coefficients were very high at 0.98 and 0.92 in the global mean for $CO_2$ and methane, respectively, with overall magnitudes consistent. We therefore conclude that MAGICC is able to reproduce the general isolated greenhouse gas forcing results (temporal patterns and magnitudes) of a more sophisticated coupled global climate model, providing confidence in the use of MAGICC for understanding the climate implications of methane mitigation analyses.

Further, we find that methane accounts for a considerable fraction of 20th Century and early 21st Century warming—roughly

half that of $CO_2$'s warming response. However, there are some features present in CM3 results without parallels in MAGICC. The features are, however, consistent in magnitude with forcing and temperature fluctuations due to unforced variability, and therefore are unable to be classified as robust responses. This highlights how unforced variability present in sophisticated models can make it difficult to ascertain robust responses to small changes in multiple forcings individually, further justifying the use of a model such as MAGICC beyond pure accessibility. To overcome this challenge, a larger number of ensembles

could be employed or simulations can be run with a quasi-chemistry-transport model (Deckert et al. 2011).

Overall, we find that reduced complexity climate models, with the MAGICC model as an example, are able to satisfactorily match the global mean temperature response to increases in isolated greenhouses gases as simulated by the GFDL-CM3, a

complex chemistry-climate model. Furthermore, we find that the prominent role of unforced variability in AM3 and CM3 makes it difficult to clearly assess climate responses to small forcing changes, ultimately supporting further use of models like MAGICC, that have little to no unforced variability, for analysing climate responses to future changes in methane emissions.

**Code availability**

5 The MAGICC v6 model is available for download at: http://www.magicc.org/download upon registration. Only the executable file is available, and not the source code. The user manual can be accessed at: http://wiki.magicc.org/index.php?title=Manual_MAGICC6_Executable. Full model details along with nineteen sets of AOGCM-calibrated parameters used here for the physics-driven ensemble members are found in Meinshausen et al. (2011a). We update the default values of methane and tropospheric ozone radiative efficiency and methane atmospheric lifetime to 10 values in Myhre et al. (2013).

The atmospheric model component (AM3) source code for GFDL CM3 is available here: https://www.gfdl.noaa.gov/am3-model/. The ocean model component (MOPM5) source code for GFDL CM3 is available here: https://www.gfdl.noaa.gov/mom-ocean-model/.

**Data availability**

15 Results from CM3/AM3 simulations and from the MAGICC model are available from Vaishali Naik (vaishali.naik@noaa.gov) and Ilissa Ocko (icko@edf.org), respectively, upon request.

**Acknowledgements**

Ilissa B. Ocko was funded by the Robertson Foundation and Heising-Simons Foundation. We thank Larry W. Horowitz for performing the long control simulation of CM3, and Alexandra Jones, Michael Winton, and Steven Hamburg for reviewing 20 our manuscript.

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

| | Experiments | Abbreviation | MAGICC v6 | GFDL CM3 |
|---|---|---|---|---|
| | All-Forcing | AllForc | X | X |
| | $CO_2$ concentrations held constant at 1860 levels | $CO_2$1860 | X | X |
| | Methane concentrations held constant at 1860 levels | $CH_4$1860 | X | X |
| Direct Simulations | All-Forcing with methane chemistry turned off | $CH_4$nochem | X | |
| | Methane concentrations held at 1860 levels with methane chemistry turned off | $CH_4$1860nochem | X | |
| | Methane concentrations held at 1860 for radiation | $CH_4$1860chem | | X |
| | Methane concentrations held at 1860 for chemistry | $CH_4$1860chem | | X |
| Derived Simulations | $CO_2$-only | | AllForc – $CO_2$1860 | |
| | $CH_4$-only | | AllForc – $CH_4$1860 | |
| | $CH_4$-direct | | $CH_4$nochem – $CH_4$1860nochem | AllForc – $CH_4$1860rad |
| | $CH_4$-indirect | | $CH_4$-only – $CH_4$-direct | AllForc – $CH_4$1860chem |

**Table 1. Direct experiments performed by MAGICC and GFDL-CM3 models, as well as derived simulations. All-Forcing simulations include time-varying natural and anthropogenic forcings but land-use held constant. Each experiment is run for 19 physics-driven ensemble members for MAGICC and three initial condition-driven ensemble members for GFDL-CM3 over the period 1860 – 2014.**

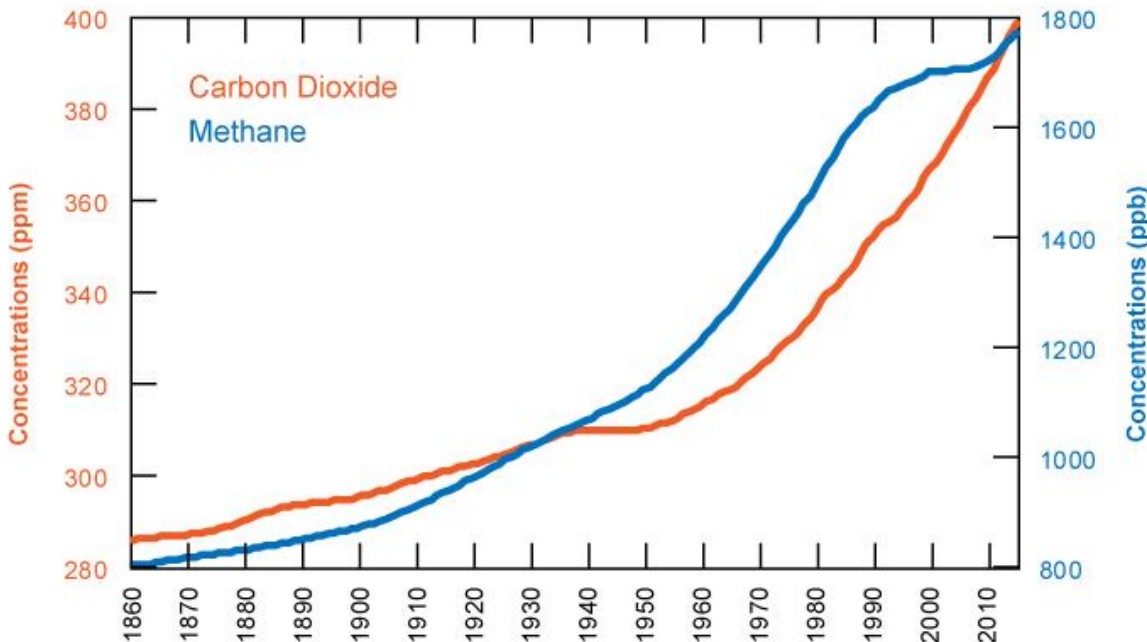

**Figure 1. Atmospheric concentrations of carbon dioxide in parts per million (orange) and methane in parts per billion (blue) used in this study (Meinshausen et al. 2011b). Note that concentrations are prescribed for CM3 throughout this time period, but only prescribed for MAGICC through 2005, of which methane emissions inputs drive the model from 2006-2014. The resulting concentrations are plotted here.**

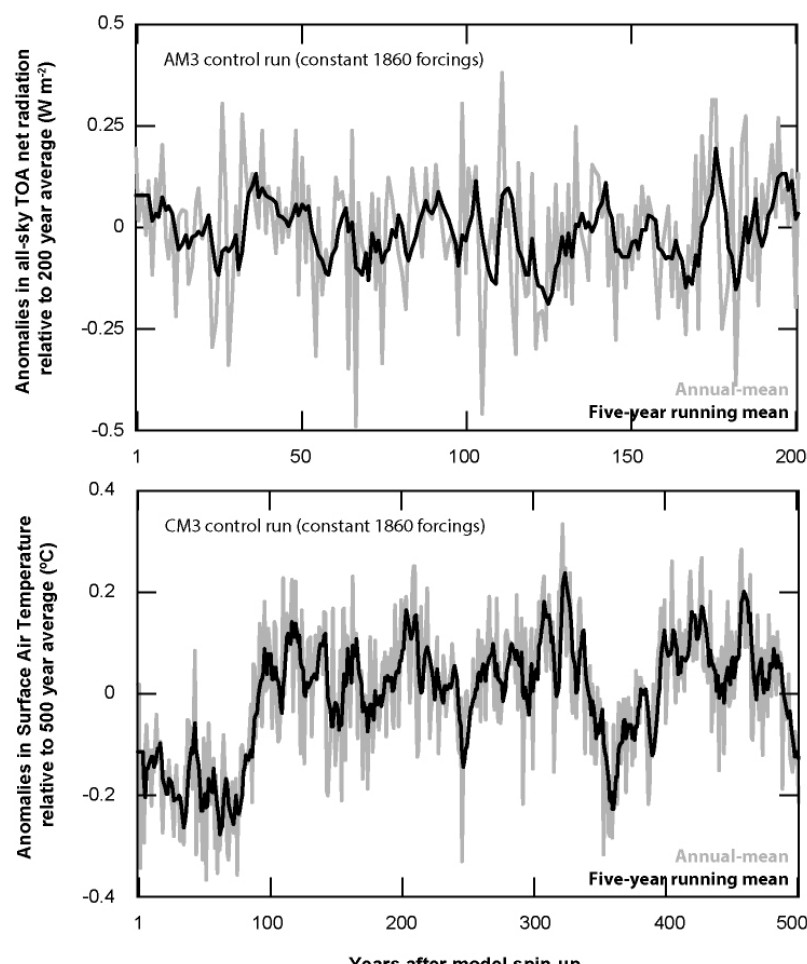

**Figure 2. Anomalies in (a) all-sky top-of-the-atmosphere (TOA) net radiation (Wm$^{-2}$) with respect to 200 year mean and (b) surface air temperature (ºC ) with respect to 500 year mean of the control simulation of AM3 and CM3 with preindustrial (1860) forcings held constant post spin-up, respectively.  Fluctuations indicate unforced variability. Results are shown for annual averages (grey line) and five-year running means (black line).**

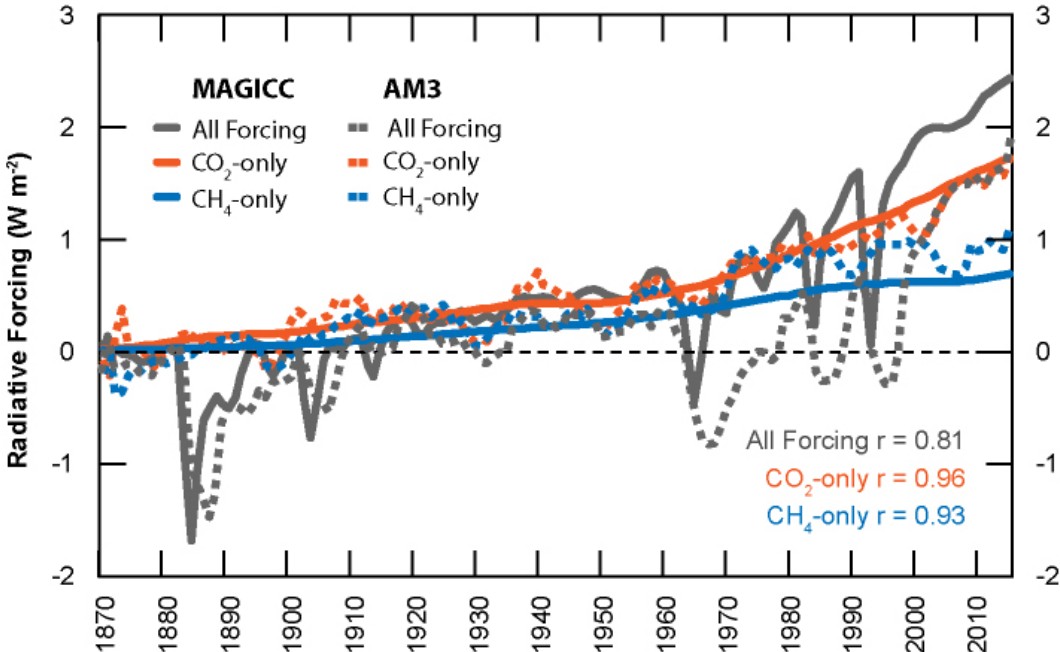

**Figure 3. Radiative forcings (W m$^{-2}$) after stratospheric adjustment due to all forcing (grey), CO$_2$-only (orange), methane-only (blue), for both AM3 (dashed) and MAGICC (solid) model simulations. Methane forcing includes its direct as well as indirect effect from influences on chemistry. AM3 radiative forcings are 'effective' radiative forcings (ERF), and include tropospheric adjustments as well, and are calculated at the top-of-atmosphere (TOA). MAGICC radiative forcings are calculated at the tropopause. AM3 data are 5-year running means. Correlation coefficients between MAGICC and AM3 forcings are shown inset.**

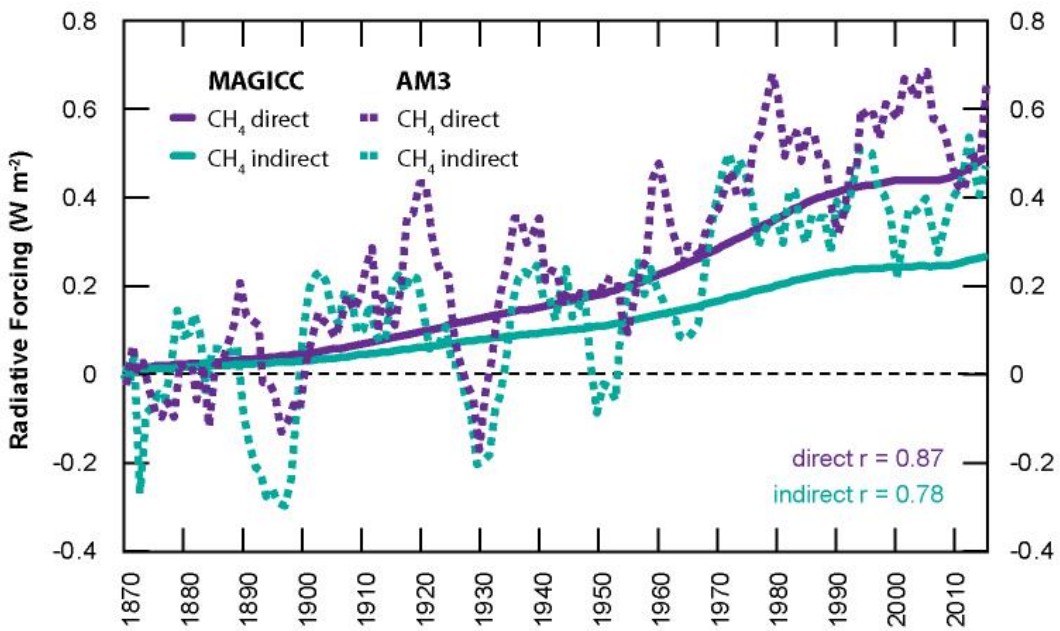

**Figure 4. Same as Figure 3, but showing direct (purple) and indirect (from methane's influence on ozone and water vapour, green) forcings (W m⁻²)**

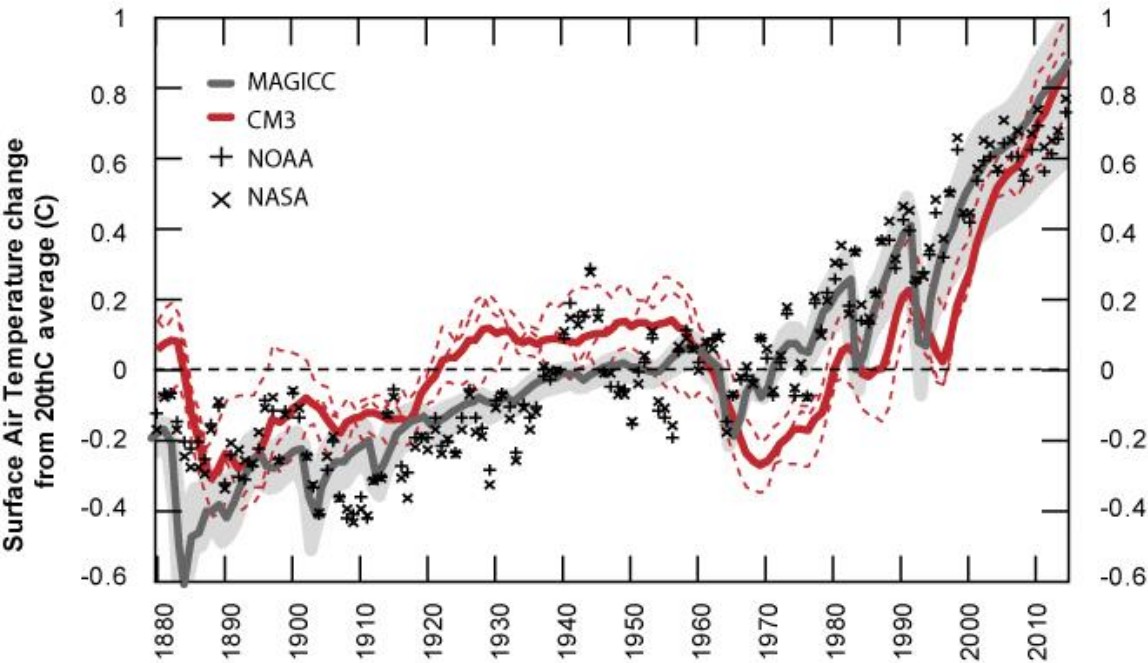

**Figure 5. All forcing global-mean surface air temperature responses in ºC for CM3 (solid red line) and MAGICC (solid grey line) model simulations as compared to observations by NOAA (+) (https://data.giss.nasa.gov/gistemp/) and NASA (x) (https://www.ncdc.noaa.gov/cag/time-series/global). All annual temperature anomalies shown as change from 20ᵗʰ Century average for each dataset. Individual initial condition-driven ensemble members for CM3 runs shown in thin dashed red lines. Physics-driven ensemble-member range for MAGICC shown as shaded grey. CM3 data are 5-year running means.**

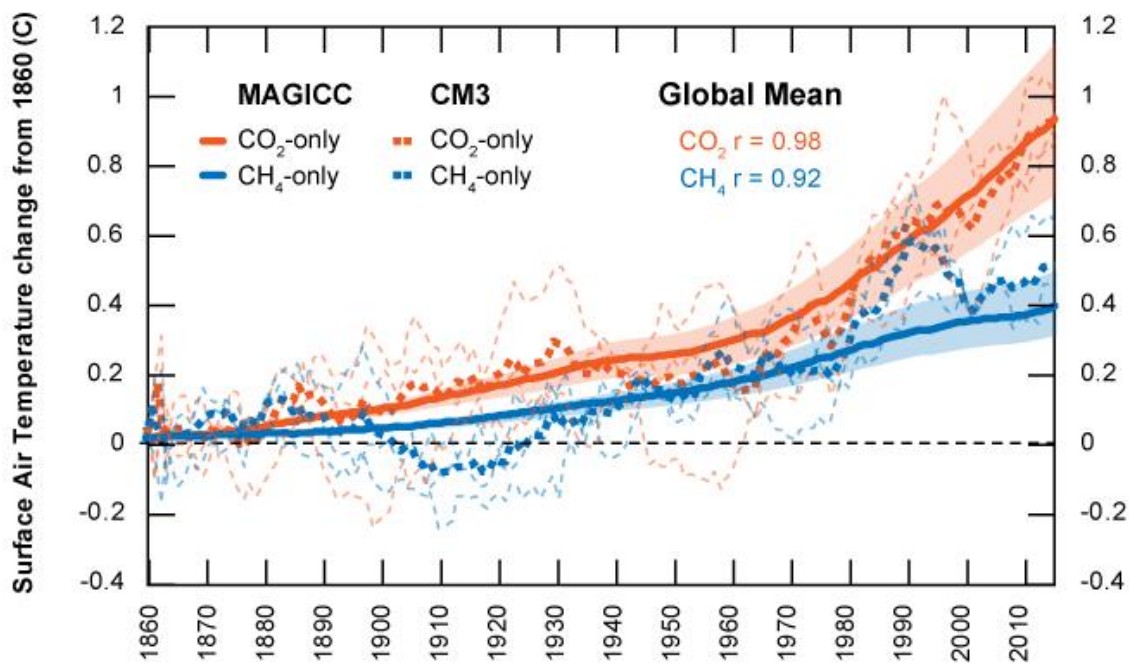

**Figure 6. Global mean surface air temperature responses in ºC for CM3 (dashed line) and MAGICC (solid line) model derived simulations – CO₂-only (orange) and methane-only (blue). Individual initial condition-driven ensemble members for CM3 runs shown in thin dashed lines. Range for MAGICC physics-driven ensemble members shown in shaded colours. CM3 data are 5-year running means. Correlation coefficient between MAGICC and CM3 temperature responses are also shown.**

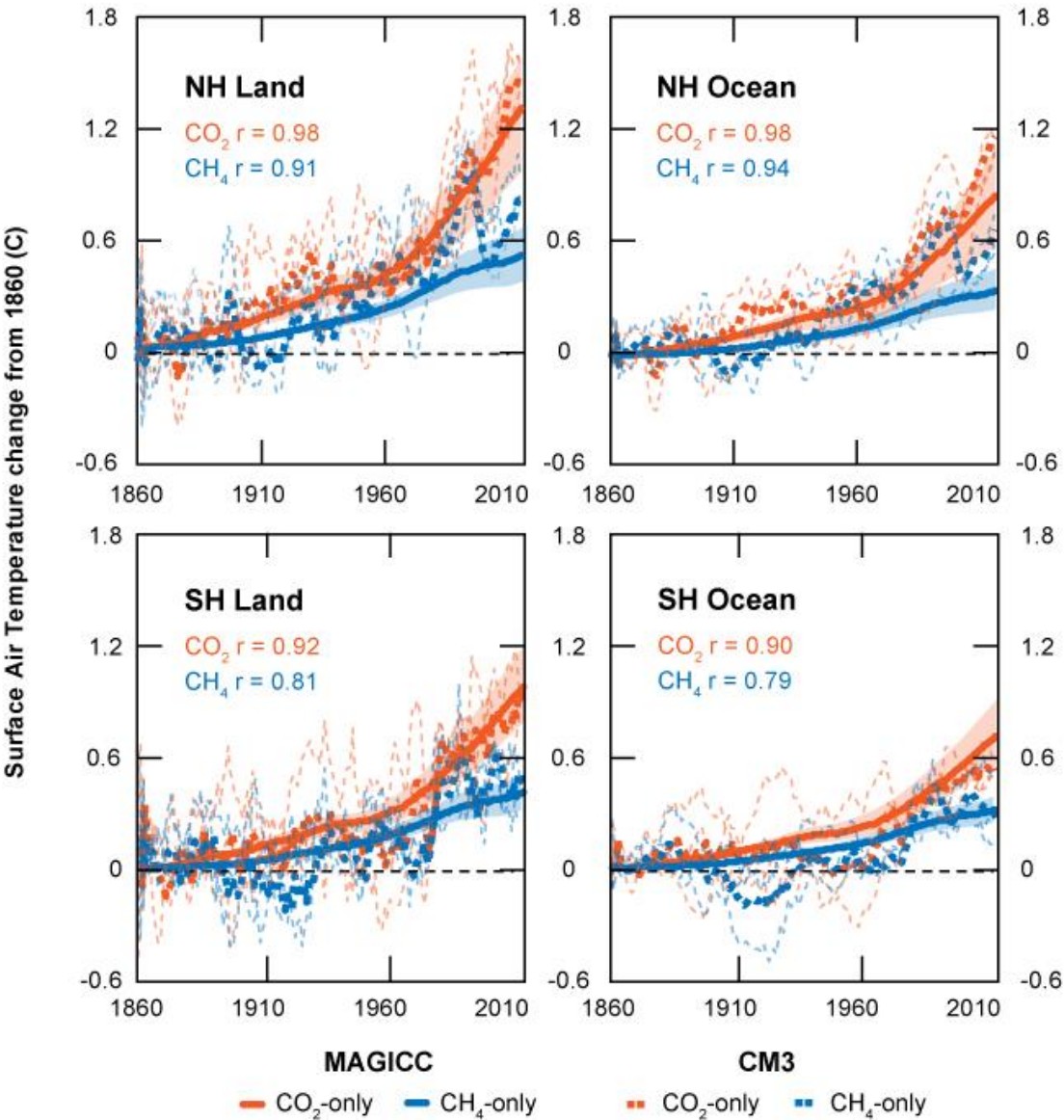

**Figure 7. Regional surface air temperature responses in ºC for CM3 (dashed line) and MAGICC (solid line) model indirect simulations – CO₂-only (orange) and methane-only (blue). Individual initial condition-driven ensemble members for CM3 runs shown in thin dashed lines. Range for MAGICC physics-driven ensemble members shown in shaded colours. CM3 data are 5-year running means. Correlation coefficient between MAGICC and CM3 temperature responses are also shown.**