# Peer review of "Rapid and reliable assessment of methane impacts on climate"

_Atmospheric Chemistry and Physics, 2018_

## Referee Comment (RC1) · Anonymous Referee #1 · 6 Feb 2018

This is a well-motivated paper examining the response of simple and advanced climate models to methane forcing and reflects work carried out in a major and innovative study. In its present form, however, it has a number of problems that might be overcome by a substantial revision. At heart, I ask whether an alternative title (and hence perspective) for this paper could be something along the lines of "The difficulty of using small ensembles of simulations of an ESM with large interannual variability to validate simple climate models for cases of small forcings". Such a perspective would still lead to a valuable publication.

General comments

1. Too much of the paper is written in language that implies that the CM (or AM3) is validating MAGICC. While this is a natural assumption for large forcing (or large

ensemble) cases, I am not sure this is necessarily the case here. The issue is that the overall transient response to historical methane forcing is around 0.5 deg C and the unforced variability in CM3 (Figure 7 and line 10-15) shows persistent and rapid unforced changes of this magnitude are possible. From a structural point of view, I feel that Figure 7 should be one of the first figures in the paper, rather than the last, where it appears as almost an afterthought. It should be accompanied by clear statement that there is a major difficulty in using ESMs to validate simple models in cases where the forcing is much less than, say, 1 W/sq.m, especially in ESMs where interannual variability is high. While the text in the intro and conclusions says that one issue is that CCMs are generally inaccessible, it could also be argued on the basis of this paper that they are potentially inappropriate tools when discussing small forcing scenarios. The issue is raised briefly at line 2-20 but the text never really returns to it when CM3 results are discussed.

2. I have a related concern about the (unforced) radiation budget variability in AM3, and how this looks compared to available observations. I note "unforced variability" in AM3 is briefly mentioned at 7-28 and 8-11, but it is unclear whether this is a major or minor issue. This is important to know for understanding the reliability of forcings derived from AM3 simulations (in Figure 2 and 3) - there are several surprising features in those forcings which may have their origin in this variability. These include the fact (line 8-9) of the AM3 methane forcing being close to that of CO2 before 2000, and exceeded it in the 1970s (which, if substantiated, would constitute a major result) and the fact that (line 8-12) AM3 methane forcing decreases when methane is flat (how can that be?) and others referred to below. I am a bit suspicious as to how and why the AM3 methane and CO2 forcings diverge after 1995 after being so close to each other before this and would like to see a clearer explanation on page 8. See also my main comment on 7-25.

3. The lack of clarity about the large unforced variability in CM3 until near the end means that some of the earlier discussion attempts a too deterministic analysis of

CM3-MAGICC differences.

Specific comments

4-26: The paper uses two quite different meanings of ensemble but does so in an interchangeable way. At 4-19, the MAGICC ensemble could be considered a perturbed-physics one. At 6-24 the CM3 ensembles are initial condition ensembles. This major distinction gets lost later, for example, in Fig 5 caption, but I have other concerns. Is a 3-member ensemble CM3 considered enough given the size of the applied forcing (and the size of this model's unforced variability)? While I appreciate the point made at 6-19 to 6-22, would it be useful to highlight the MAGICC configuration that matches CM3 as closely as possible, particularly for climate sensitivity? As far as I can tell (and sorry if I miss it) the reader is never told what the climate sensitivity of CM3 is, and therefore it is hard to judge whether it might be an outlier in the MAGICC ensemble. Finally, I presume the AM3 results were not from an ensemble?

7-25 and Figure 2: I understand that stratospheric aerosol RF is not a topic for this paper, but the difference between MAGICC and AM3 is very striking and it seems that the AM3 forcing from individual volcanoes is persisting for unrealistic lengths of times (most notably Agung) and leaves a signature which looks unrealistic in the surface temperatures (in Figure 4). Has this been discussed in earlier papers? Is it some lingering indirect effect of the eruptions? Is it due to the smoothing process? I feel it should be mentioned briefly here, not least because it may impact on the ability to extract the methane signal from CM3.

8-14 and Figure 3: There are quite a few concerning features in this figure, which leads on from my other comments. Given the smoothly varying methane concentration (Figure 1) how can the direct AM3 forcing fluctuate so rapidly? Is this just the unforced variability of the radiation budget in AM3? Also it seems that there is a strong degree of anticorrelation between the AM3 direct and indirect forcings most apparent in some periods (1920-1940 and 1980 onwards) which lead to the total forcing (in Figure 2)

being much smoother – so again, is this anticorrelation just an artefact of the analysis method, or robust? Finally, the fact that the methane indirect forcing falls to near zero in the 2000's would, if correct, constitute a major result. The text (8-20) says that it is "mostly tropospheric ozone" but this needs substantiating by showing some measure of the tropospheric ozone variations in AM3, and some assurance that the feature is a clear signal above the noise.

8-6 to 8-13: I have quite a few comments on this paragraph. (i) I think comparison of MAGICC with IPCC needs to be more carefully done. I am almost certain they both use the same radiative forcing expressions and hence the only reasons for differences would come from different scenarios of trace gas change, and possibly different handling of N2O overlap. (ii) wording like "MAGICC reasonably reproduces . . . AM3" (8-6) and "not captured by MAGICC" (8-11) implies to me that there is a belief that the AM3 radiation code is superior to the radiative transfer codes used to generate the RF expressions in MAGICC and IPCC. Of course, this may be the case, but if this cannot be established with certainty, the wording should be more cautious. And as noted in my general comment (2) without knowing the size of the unforced variability in AM3 it is hard to place any differences in perspective.

9-13: It is hard to know what conclusions to draw from this paragraph – is it that the CM3 signals are unreliable because of the difficulty of extracting a small signal from a small ensemble given the size of the unforced variability (especially in the 1900-1915 period when methane forcing was less than 0.2 W m-2)?

10-10 and especially 10-13: Again it is hard to know what conclusion to draw here, and there is nothing to lead the reader. I believe that this paragraph should come much earlier in the paper and be flagged as a major caveat when trying to extract small signals from ESMs, and in particular CM3 given its such large variability. As it is, this aspect is presented almost as an afterthought.

11-12: "useful threshold" – I agree with this, but I don't believe the paper applies this

threshold. If it did, I think one conclusion could be that some of the early 20th century signals attributed to methane in AM3/CM3 are not robust and should not be interpreted as such. This may be what 11-13 to 11-15 is trying to say. If so, it needs to be stated more clearly, earlier in the paper and reflected in the abstract.

Additional comments

Throughout: I suggest that the 4 in CH4 is subscripted throughout, for consistency with CO2.

Throughout: There is a lack of consistency in the labelling of simulations – compare the abbreviation column of Table 1 with Equations (1) to (4).

1-7: The abstract does not highlight any of the scientific results emerging from CM3 – the fact that methane-induced warming is competitive with CO2 for long periods and the high variability of its indirect effect. My other comments indicate some scepticism about these results, but it seems odd not to mention them at all in the abstract if the authors stand by them.

2-3: I suggest this sentence is re-worded to avoid the appearance of policy advocacy. The standard and effective IPCC wording is "if we want to avoid warming then we need to reduce emissions".

2-10: I looked at Etminan's paper and could see nothing on temperature change.

2-16: For sure, the GWP concept has a lot of problems, but the strong variation of its value with time horizon (e.g. compare GWP(20) with GWP(100)) does capture "important temporal distinctions".

4-1 and 4-7: Is it correct that MAGICC is driven by concentrations for 1765-2005 and emissions for 2006-2014? The fact that the methane lifetime is mentioned as being updated at 6-15 adds to confusion as to whether the model is concentration or emission driven. The (possible) emission/concentration confusion is also present in the paragraph after 6-15. But then 7-17 and Figure 1 seems to clearly imply that both models

are concentration driven. So I ended up a bit confused.

4-15: There are significant differences in the CO2 expression used in the two references given here – I doubt that MAGICC uses the IPCC FAR expression.

4-25: Units of climate sensitivity missing.

5-19: A minor query, but I wasn't sure what the "indirect feedbacks of CH4 on CO2" were – it could include the fact that CH4 is oxidised to CO2, or that CH4-induced climate change impacts the carbon cycle or CH4-induced changes in O3 impact the carbon cycle by their effect on vegetation (or all three).

6-15: Coupled with my comment at 4-15, I am confused. Does MAGICC use the AR5 forcing expressions for methane or does it use the radiative efficiencies (which are W m-2 ppbv-1 normally defined for small perturbations from present day)? I don't think MAGICC can use both and this needs to be clarified.

7-6: I found this sentence a bit cryptic. Presumably N2O and its overlap are considered in the AM3 radiation code? I was not sure why the dependence should lead to a "likely overestimate" of methane forcing.

8-28: I don't agree with NASA and NOAA temperature series being labelled as "independent observational datasets" when there is, of course, a huge commonality in the underlying data, and I believe in some of the applied corrections to that data. A better wording could be used.

11-1: "accurately" – the word "adequately" is used at 9-1 and I believe it a much better description of Figure 4.

11-21: One issue with MAGICC is that (as I understand it) only the executable, rather than the source code, is available, which is at odds with the title at 11-19. It may be worth making this clear to readers.

18-4: Delete "technically"?

---

## Short Comment (SC1) · 24 Apr 2018

I agree with the authors about the value of simple analytical tools for quantifying the physical impact of individual greenhouse gas mitigation options, and see this paper as a good contribution to the literature of evaluating how these simple tools might compare to more complex ones. I have a couple of comments for the authors' consideration.

1) MAGICC is, of course, one of the most widely used tools for this purpose, and therefore a reasonable choice. However, it might be worthwhile for the authors to discuss a couple of alternatives.

a. Hector is in a similar class of model as MAGICC, but has the advantage of being fully open-source (see comment on 11-21 from Referee 1). Hector

is described in Hartin et al. 2014, https://core.ac.uk/download/pdf/25503085.pdf. Also relevant is a thesis by Schwarber on comparing Hector and MAGICC at https://www.atmos.umd.edu/theses_archive/2016/aschwarber_masters.pdf. I am not suggesting running Hector for this paper, as that would be a large lift, but a citation and brief mention of the benefits of open source could be worthwhile.

b. Meanwhile, there are approaches that are even simpler than MAGICC. Melvin et al. (2016), for example, estimated the physical impacts resulting from methane mitigation by using the simplified expressions from AR5 for concentration and radiative forcing and from Shine et al. (2005) for temperature. It could be a valuable sensitivity analysis to take one of these simplified approaches as an additional comparison. These simplified equations may be better suited to analyzing the marginal effect of perturbations in emissions, rather than to simulate overall global temperature change from total anthropogenic emissions: however, it seems to me that this kind of marginal analysis is consistent with the goals of this paper.

c. The NAS in their report on valuing climate damages (https://www.nap.edu/catalog/24651/valuing-climate-damages-updating-estimation-of-the-social-cost-of) suggest the use of the FAIR model: this would fall between Hector/MAGICC and the GTP-style equations in terms of complexity.

2) As with Referee 1, I find it surprising that the AM3 forcing results only diverge in the last couple decades in contrast with MAGICC which shows a slowly growing divergence over the entire run. Would there be any effect of running AM3 with different initial conditions, which could show whether this is a robust result or a result deriving from internal variability? The only other explanation that comes to mind other than initial condition sensitivity is that somehow CO2 and CH4 forcing have differential sensitivity to SSTs or sea-ice extent. Maybe a constant-concentration experiment could be informative in terms of whether the forcing of methane and CO2 might respond differently to the historical changes of SST & sea-ice?

3) I do find this comparison of more complex models to simple models to be an informative exercise (see Sarofim 2012 where I used both the MIT IGSM and MAGICC to calculate the 100-year sustained GTP for methane: https://link.springer.com/content/pdf/10.1007%2Fs10666-011-9287-x.pdf). But, as Referee 1 notes, this comparison is complicated by the variability inherent in more complex models, even as at the same time, this is one of the motivators behind the use of simple models for investigating the effect of emissions perturbations that are expected to have temperature effects smaller than the internal variability of the complex models. I don't have a good answer for this, other than averaging even larger ensembles in order to reduce initial-condition-based noise even more.

---

## Referee Comment (RC2) · Anonymous Referee #2 · 25 Apr 2018

The paper shows the importance of surrogate models in order to compare the impact of different climate species with low computational effort and tries to evaluate the open source surrogate model MAGICC by comparing the temperature responses to historical methane and CO2 emissions from MAGICC and GFDL CM3. This is an important work as surrogate models are very useful for analysing mitigation scenarios as they are computational effective and can assess small forcings. Nevertheless it is difficult to evaluate surrogate models with small forcings with a complex model with large internal variability.

Specific comments

1. Simplified models have beside the lower need of computational resources the advantage that the internal variability is small or zero and it is possible to assess the

impact of small changes, while the internal variability in complex models is too large therefore. But this makes it at the same time difficult to compare them and evaluate the simplified model. As Reviewer #1 stated it is difficult to evaluate forcings which are in the same order as the unforced internal variability (Fig 7) and the variation of different ensemble members (e.g. Fig. 5, 1960). The fact that the internal variability of CM3 is very large compared to CMIP5 models, should be mentioned earlier in the text to make it easier to put the results in the right context. The text is partly formulated as CM3 is the truth and MAGICC should reproduce the same features. While this is important if the features are physically base, it is not the case if the features are due to internal variability, as the benefit of simplified models is that the results are almost free of internal variability. Additionally Fig 4 suggests that MAGICC provides better agreement with observations than CM3 does. Similar to reviewer #1 I would suggest putting more focus on the fact that it is difficult to evaluate simplified models by complex models with large variability. In addition some possible ways to overcome this problem could be provided, e.g. larger number of ensembles or simulations with a quasi-chemistry-transport model mode (e.g. Deckert et al., 2011).

2. For my opinion the description of the models and simulations should be more detailed. I had for example some difficulties to exactly understand what the models use as an input and which parameter were calculated by the models.

- Are the concentrations (p4-l1) or the emissions (p4-l7) prescribed in MAGICC?

- Was the choice of the ensemble members of CM3 randomly or did you choose years with extreme or mean values?

- How is the RF calculated in CM3?

- Why does All Forcing in MAGICC have a large variability, while the $CO_2$ and $CH_4$ do not have one? Are the forcings (except $CO_2$ and $CH_4$) prescribed?

- Why does $CO_2$ show negative Forcing in Fig 2 although the concentration increases?

- Why are direct and indirect CH4 effects anti-correlated or have a time lag? Is there a physical explanation or is it an artifact of the internal variability?

- Why is the temperature change of CH4 of CM3 negative although the forcing is positive?

p6-l5 MAGICC simulates from 1750-2100, but in p4-l1-9 only information about concentrations and forcings between 1765 and 2014 are given

P6 l21 Does the 'downloaded' version of MAGICC include tuning to the multi-model-mean or can be chosen which AOGCM is used for calibration?

P8-25 A description about the kind of data used should be included

Technical comments

P5-l12 comma is missing after carbon dioxide

P6-29 Here RF is defined at the tropopause, while it is defined at the top of the atmosphere in p7-23

P8-3 change 'slightly offset' in 'offset' (1W/m2 is large compared to the forcing)

P11-1 change 'accurately' in 'adequately'

Is there a reason why the Fig starts in different years (1860, 1870 or 1880)?

Publikation: Deckert, Rudolf und Jöckel, Patrick und Grewe, Volker und Gottschaldt, Klaus-Dirk und Hoor, Peter (2011) A quasi chemistry-transport model mode for EMAC. Geoscientific Model Development, 4, Seiten 195-206. Copernicus Publcations. DOI: 10.5194/gmd-4-195-2011 ISSN 1991-959X
* * *

---

## Author Comment (AC1) · 23 Jul 2018

We sincerely appreciate the careful reviews and helpful suggestions provided by the Reviewers, and thank the Reviewers and the Editor for their time. We have made several changes to the manuscript in response to the comments that have considerably enhanced the manuscript.

Major changes to the paper include: analysis and discussion of unforced variability in AM3/CM3 expanded and moved to the beginning of the results section rather than the end; an additional control run simulation performed to isolate unforced variability in AM3, and the results added to the figure with the CM3 control run results; more emphasis on unforced variability as an additional reason why simplified models are

preferred tools for impacts of small changes; text modifications to caveat the difficulty in comparing simple model results with that from more complex models; inclusion of strategies to overcome the challenge of unforced variability; additional simulations performed using AM3 and CM3 to provide a different method for calculating the indirect (via chemistry) forcing and response due to historical methane changes; inclusion of reduced-complexity climate model options as alternatives to MAGICC; text modifications to reduce the impression that CM3 is "truth;" and addition of 14 new references.

In the attached supplemental document, we have responded point-by-point to comments (reviewer comments in blue, authors' responses in black), and have included the revisions in the text with and without tracked-changes.

Please also note the supplement to this comment:
https://www.atmos-chem-phys-discuss.net/acp-2018-26/acp-2018-26-AC1-supplement.pdf

**Supplement:**

**Manuscript Ref: acp-2018-26**

**Rapid and reliable assessment of methane impacts on climate**

Ilissa B. Ocko, Vaishali Naik, and David Paynter

We sincerely appreciate the careful reviews and helpful suggestions provided by the Reviewers, and thank the Reviewers and the Editor for their time. We have made several changes to the manuscript in response to the comments that have considerably enhanced the manuscript. Below, we have provided information on the major modifications to the text and responded point-by-point to comments (reviewer comments in blue, responses in black).

Major changes to the paper include:
- analysis and discussion of unforced variability in AM3/CM3 expanded and moved to the beginning of the results section rather than the end;
- an additional control run simulation performed to isolate unforced variability in AM3, and the results added to the figure with the CM3 control run results;
- more emphasis on unforced variability as an additional reason why simplified models are preferred tools for impacts of small changes;
- text modifications to caveat the difficulty in comparing simple model results with that from more complex models;
- inclusion of strategies to overcome the challenge of unforced variability;
- additional simulations performed using AM3 and CM3 to provide a different method for calculating the indirect (via chemistry) forcing and response due to historical methane changes;
- inclusion of reduced-complexity climate model options as alternatives to MAGICC;
- text modifications to reduce the impression that CM3 is "truth;" and
- addition of 14 new references.

**Responses to RC1 (Anonymous Referee #1):**

**Comment 1:** Too much of the paper is written in language that implies that the CM (or AM3) is validating MAGICC. While this is a natural assumption for large forcing (or large ensemble) cases, I am not sure this is necessarily the case here. The issue is that the overall transient response to historical methane forcing is around 0.5 deg C and the unforced variability in CM3 (Figure 7 and line 10-15) shows persistent and rapid unforced changes of this magnitude are possible. From a structural point of view, I feel that Figure 7 should be one of the first figures in the paper, rather than the last, where it appears as almost an afterthought. It should be accompanied by clear statement that there is a major difficulty in using ESMs to validate simple models in cases where the forcing is much less than, say, 1 W/sq.m, especially in ESMs where interannual variability is high. While the text in the intro and conclusions says that one issue is that CCMs are generally inaccessible, it could also be argued on the basis of this paper that they are potentially inappropriate tools when discussing small forcing scenarios. The issue is raised briefly at line 2-20 but the text never really returns to it when CM3 results are discussed.

> **Response:** We thank the referee for the constructive feedback, and have made a number of changes in response to the comments.
>
> We have reorganized the paper based on the referee's suggestion to include Fig. 7 as one of the first figures, and have considerably expanded analysis and discussion of unforced variability early in the paper and throughout the text. We have also run an additional simulation to assess AM3 unforced variability in addition to CM3, and added the results to the previous Fig. 7.
>
> The reason that the paper was written as the CCM (AM3/CM3) validating MAGICC is because the CCM has a much more sophisticated treatment of relevant chemistry and physics, and therefore could be argued as more advanced in modeling climate responses to methane forcings—especially because it can resolve responses on regional scales and account for the spatially heterogenous response from chemical changes. However, because of the unforced variability in CCMs, we agree that an additional reason to use MAGICC is that it does not include unforced variability and therefore can more clearly address changes in small forcing scenarios, and have expanded the discussions in the text to reflect this—including acknowledgement in the abstract as both a motivator and result of our study. We have also toned down the language that implies that CM3 is "truth" in this analysis.

**Comment 2:** I have a related concern about the (unforced) radiation budget variability in AM3, and how this looks compared to available observations. I note "unforced variability" in AM3 is briefly mentioned at 7-28 and 8-11, but it is unclear whether this is a major or minor issue. This is important to know for understanding the reliability of forcings derived from AM3 simulations (in Figure 2 and 3) - there are several surprising

features in those forcings which may have their origin in this variability. These include the fact (line 8-9) of the AM3 methane forcing being close to that of CO2 before 2000, and exceeded it in the 1970s (which, if substantiated, would constitute a major result) and the fact that (line 8-12) AM3 methane forcing decreases when methane is flat (how can that be?) and others referred to below. I am a bit suspicious as to how and why the AM3 methane and CO2 forcings diverge after 1995 after being so close to each other before this and would like to see a clearer explanation on page 8. See also my main comment on 7-25.

**Response:** We thank the reviewer for the thoughtful comments. In order to evaluate the unforced variability in AM3, we have run AM3 with 1860 forcings for 200 years with a repeating seasonal cycle of sea surface temperatures and sea ice for each year (see figure below of a time series of net radiation at the top-of-atmosphere for all-sky conditions for annual and 5-year running mean averages, which has also been added to the new Figure 2 in the paper). We find that based on unforced variability, one year averages can easily have swings of 0.8 Wm-2, while 5-year averages is around 0.35 Wm-2. Ten-year averages is better at ~0.20 Wm-2.

Comparing this run to the CO2-only and methane-only AM3 forcings (5-yr averages) and using the MAGICC forcings as a baseline, we find that all of the deviations of CO2-only AM3 forcings, and nearly all of the deviations of methane-only AM3 forcings, fall within 0.35 W/m2 from the baseline. Therefore, it is very likely that the unforced variability in CM3 is the source of these variations, and this is another reason why a reduced-complexity model unfettered by unforced variability is useful. We have clarified both of these points in the text (lines 10-12 – 10-15): "*Some unexpected features in AM3 forcings (such as negative forcings in the earlier years despite increasing atmospheric concentrations) are likely due to unforced variability. Using the MAGICC forcings as a benchmark for a signal due to forced changes only, we find that nearly all of the deviations of AM3 fall within the range of internal variability as derived from the control simulation: 0.35 W m-2.*"

However, we note that the AM3 methane and CO2 forcings diverging after 1995 is consistent with the leveling off of atmospheric methane concentrations around the mid-90s and through the mid-2000s (see Figure 1 in the paper) and this is acknowledge in the paper.

[Figure]

**Comment 3:** The lack of clarity about the large unforced variability in CM3 until near the end means that some of the earlier discussion attempts a too deterministic analysis of CM3-MAGICC differences.

> **Response:** We appreciate this observation, and have reorganized the discussion to address unforced variability in AM3 and CM3 in the beginning of the Results section, considerably expanding the text and analysis, and have referred back to this variability throughout the rest of the text.
>
> An example of the reframed discussion include (lines 9-20 – 10-5): "*Given that an important difference between AM3/CM3 and MAGICC results is the role of unforced variability in AM3/CM3, we first analyse the magnitudes of unforced variability in both AM3 and CM3. Although initial condition-driven ensemble member means and/or running averages are employed to dampen out some of the variability in AM3/CM3, it still plays a large role in forcing and temperature responses. CM3, in particular, has been shown to produce magnitudes of variability on the upper end of CMIP5 models (Brown et al., 2015).*
>
> *The results of the control simulations with constant preindustrial (1860) external radiative forcings are shown in Fig. 2. In the case of AM3 unforced radiative forcings at the top-of-atmosphere for all-sky conditions range from -0.18 to 0.21 W m-2 with a standard deviation of 0.07 W m-2 for a five-year running mean. We find the maximum swing between two consecutive five-year means to be 0.35 W m-2. Sources of unforced variability in AM3 include a mixture of land snow/ice cover variability and just year-to-year variability in the weather; soil moisture may also play a role. For CM3, unforced internal dynamics yield temperature responses ranging from -0.27 to 0.24 °C for five-year running means with a standard deviation of 0.1 °C. We find the maximum swing between two consecutive five-year means to be 0.2 °C. The variability is driven by interactions among the ocean-atmosphere-land systems. While unforced variability is a key component to modelling the climate system, it can mask or amplify responses to*

*external forcings over short timescales (e.g., Brown et al., 2017). This makes it difficult to clearly assess responses to small external forcings, and provides further motivation for using simpler models like MAGICC for analysis of small forcing scenarios."*

**Comment 4:** Specific comments:

4-26: The paper uses two quite different meanings of ensemble but does so in an interchangeable way. At 4-19, the MAGICC ensemble could be considered a perturbed physics one. At 6-24 the CM3 ensembles are initial condition ensembles. This major distinction gets lost later, for example, in Fig 5 caption, but I have other concerns. Is a 3-member ensemble CM3 considered enough given the size of the applied forcing (and the size of this model's unforced variability)?

> **Response:** The needed clarity for ensemble definition differences is a great point by the referee. We have clarified the differences in the definitions for CM3 and MAGICC using qualifiers throughout the text and figure captions as: "initial condition-driven ensemble members" for CM3 and "physics-driven ensemble members" for MAGICC.
>
> The 3-member ensemble collection for CM3 is a standard practice for models of this scale due to limitations of computational resources. While small, studies have shown that forced changes in air temperature (as opposed to changes in atmospheric circulation and precipitation) can be detected with fewer ensemble members (Deser et al., 2010). We have now included this caveat in the text (lines 8-8 – 8-10: "*While three ensemble members is relatively small, we are limited by computational resources and studies have shown that forced changes in air temperature, as opposed to changes in atmospheric circulation and precipitation, can be detected with fewer ensemble members (Deser et al., 2012).*"
>
> Deser, C., Phillips, A., Bourdette, V. and Teng, H., 2012. Uncertainty in climate change projections: the role of internal variability. Climate dynamics, 38(3-4), pp.527-546.

While I appreciate the point made at 6-19 to 6-22, would it be useful to highlight the MAGICC configuration that matches CM3 as closely as possible, particularly for climate sensitivity? As far as I can tell (and sorry if I miss it) the reader is never told what the climate sensitivity of CM3 is, and therefore it is hard to judge whether it might be an outlier in the MAGICC ensemble.

> **Response:** As noted in the text and appreciated by the referee, the reason that we do not tune the properties of MAGICC to match that of CM3 is because we are trying to determine if MAGICC as it currently exists and is used can effectively reproduce climate changes in response to methane emissions. While tuning MAGICC to be more like CM3 would yield responses more similar to CM3, this is not how MAGICC is and will be used in the future. Here we show that based on

MAGICC's current properties it is sufficient in modeling climate responses to changes in methane emissions.

However, we appreciate the feedback of the referee in inquiring about the MAGICC configuration that matches CM3 the most. We have therefore provided information on CM3's climate sensitivity (4.8 K based on multimillenial simulations (Paynter et al., 2018)) as well as noted that two of MAGICC's physics-driven ensemble members are derived from two predecessors of CM3: CM2.0 and CM2.1.

Paynter, D., Frölicher, T.L., Horowitz, L.W. and Silvers, L.G., 2018. Equilibrium climate sensitivity obtained from multimillennial runs of two GFDL climate models. Journal of Geophysical Research: Atmospheres, 123(4), pp.1921-1941.

Finally, I presume the AM3 results were not from an ensemble?

**Response:** The referee is correct in that the AM3 results were not from an initial condition-driven ensemble, because they were driven by observed SST and SIC. We have clarified this in the text (lines 8-13 – 8-15): "*The model configuration of AM3 was exactly the same as CM3 except AM3 model integrations over the period 1870 to 2014 were performed with observed sea-surface temperature and sea-ice cover (Rayner et al., 2003), and therefore do not include an ensemble driven by different initial conditions.*"

7-25 and Figure 2: I understand that stratospheric aerosol RF is not a topic for this paper, but the difference between MAGICC and AM3 is very striking and it seems that the AM3 forcing from individual volcanoes is persisting for unrealistic lengths of times (most notably Agung) and leaves a signature which looks unrealistic in the surface temperatures (in Figure 4). Has this been discussed in earlier papers? Is it some lingering indirect effect of the eruptions? Is it due to the smoothing process? I feel it should be mentioned briefly here, not least because it may impact on the ability to extract the methane signal from CM3.

**Response:** We thank the reviewer for this observation. Upon further investigation, it appears that the "lingering" volcanic signature is mainly due to the 5-year running mean smoothing process (see figure below). We have added this caveat to the text (lines 11-31 – 12-3): "*We note, however, that the 'lingering' temperature response in CM3 to major volcanic eruptions is an artefact of the 5-year running mean smoothing process; this is why CM3 temperature responses to volcanic eruptions persist longer than what is seen in the observational records and by MAGICC. This is not found, however, to considerably impact the correlation coefficients between the CM3 data and the NOAA/NASA data.*"

CM3 simulates the impact of 'explosive' volcanic aerosols via an imposed time series of volcanic optical properties (from Stenchikov et al. (2006)) rather than direct injection of sulfur into the stratosphere (Donner et al., 2011). We have

clarified this in the text (lines 6-25 – 6-27): "*'Explosive' volcanic eruptions are imposed via a time series of volcanic optical properties rather than from direct injection of sulfur into the stratosphere (Stenchikov et al., 2006; Donner et al., 2011)."*

The response of surface temperature to volcanic forcing from CM3 is discussed in several previous papers (Austin et al., 2013; Golaz et al., 2014; Merlis et al., 2014), and what we show in our paper is similar to that shown by Golaz et al. (2014) Figure 3 (attached below), which also employs a 5-year running mean.

CM3 all-forcing temperature anomalies:

[Figure]

Golaz et al. 2013, Fig. 3:

[Figure]

**Figure 3.** Time evolution of global mean surface air temperature anomalies. Color lines represent the CMIP5 GFDL CM3 model (green) and the two alternate configurations, CM3w (red) and CM3c (blue). Each line is a five-member ensemble average. Anomalies are computed with respect to 1881–1920. Model drift is removed by subtracting from each ensemble member the linear trend of the corresponding period in the control simulation. Also shown are observations from NOAA NCDC [*Vose et al.*, 2012], NASA GISS [*Hansen et al.*, 2010], and HadCRUT3 [*Brohan et al.*, 2006]. A 5 year running mean is applied to model results and observations. Letters above the horizontal axis mark major volcanic eruptions: Krakatoa (K), Santa María (M), Agung (A), El Chichón (C), and Pinatubo (P).

Austin, J., Horowitz, L.W., Schwarzkopf, M.D., Wilson, R.J. and Levy, H., 2013. Stratospheric ozone and temperature simulated from the preindustrial era to the present day. Journal of Climate, 26(11), pp.3528-3543.

Donner, L. J., Wyman, B. L., Hemler, R. S., Horowitz, L. W., Ming, Y., Zhao, M., Golaz, J. C., Ginoux, P., Lin, S. J., Schwarzkopf, M. D. and Austin, J.: The dynamical core, physical parameterizations, and basic simulation characteristics of the atmospheric component AM3 of the GFDL global coupled model CM3, J. Clim., 24(13), 3484–3519, 2011.

Golaz, J.C., Horowitz, L.W. and Levy, H., 2013. Cloud tuning in a coupled climate model: Impact on 20th century warming. Geophysical Research Letters, 40(10), pp.2246-2251.

Merlis, T.M., Held, I.M., Stenchikov, G.L., Zeng, F. and Horowitz, L.W., 2014. Constraining transient climate sensitivity using coupled climate model simulations of volcanic eruptions. Journal of Climate, 27(20), pp.7781-7795.

Stenchikov, G., K. Hamilton, R. J. Stouffer, A. Robock, V. Ramaswamy, B. Santer, and H.-F. Graf, 2006: Arctic Oscillation response to volcanic eruptions in the IPCC AR4 climate models. J. Geophys. Res., 111, D07107, doi:10.1029/2005JD006286

8-14 and Figure 3: There are quite a few concerning features in this figure, which leads on from my other comments. Given the smoothly varying methane concentration (Figure 1) how can the direct AM3 forcing fluctuate so rapidly? Is this just the unforced variability of the radiation budget in AM3? Also it seems that there is a strong degree of anticorrelation between the AM3 direct and indirect forcings most apparent in some periods (1920-1940 and 1980 onwards) which lead to the total forcing (in Figure 2) being much smoother – so again, is this anticorrelation just an artefact of the analysis method, or robust? Finally, the fact that the methane indirect forcing falls to near zero in the 2000's would, if correct, constitute a major result. The text (8-20) says that it is "mostly tropospheric ozone" but this needs substantiating by showing some measure of the tropospheric ozone variations in AM3, and some assurance that the feature is a clear signal above the noise.

**Response:** Further investigation revealed that most of the anti-correlation and many of the 'peculiar' features was indeed an artefact of the analysis method. We ran additional simulations to calculate the indirect methane forcings and responses via modifications to chemistry rather than our original method that introduced a double subtraction to determine the indirect effects. We ran AM3 and all ensemble members of CM3 for an experiment where methane radiation varied with time but chemistry was held at 1860 methane levels. We found that this new methodology resolved many of the issues, including the majority of the anti-correlation features as well as the near-zero indirect forcing around the year 2000 (see new figure 4 in paper and below).

We have therefore updated the manuscript to reflect this new methodology for indirect effects of methane, keeping the original methodology for direct effects of methane. This method is also more consistent with how we calculated indirect effects of methane in MAGICC. While the direct and indirect effects are not completely additive (see figure below) this is typical nonlinearity effects for models of this complexity. To address the referee's comment about the ozone behavior for the different experimental setups, we plotted the below figure. This shows that total ozone column changes that we see are greater than zero due to CH4 at present-day.

Further, we note that unforced variability in AM3 does play a considerable role in the temporal evolution of forcings. We have run AM3 with 1860 forcings for 200 years with repeat sea surface temperatures and sea ice for each year (see new Figure 2). We find that based on unforced variability, one year averages can easily have swings of 0.8 Wm-2, while 5-year averages is around 0.35 Wm-2. Ten-year averages are better at ~0.20 Wm-2. Sources of unforced variability in AM3 include a mixture of land snow/ice cover variability and just year-to-year variability in the weather; soil moisture may also play a role. For indirect methane forcings in AM3, the anomalies from a MAGICC benchmark all fall within 0.25 Wm-2, which is within the realm of unforced variability. We have clarified this in the text (lines 9-25 – 9-29): "*In the case of AM3 unforced radiative forcings at the top-of-atmosphere for all-sky conditions range from -0.18 to 0.21 W m-2 with a standard deviation of 0.07 W m-2 for a five-year running mean. We find the maximum swing between two consecutive five-year means to be 0.35 W m-2. Sources of unforced variability in AM3 include a mixture of land snow/ice cover variability and just year-to-year variability in the weather; soil moisture may also play a role.*"

[Figure]

[Figure]

**Delta Total O3 Column due to Individual Forcings**

[Figure]

8-6 to 8-13: I have quite a few comments on this paragraph. (i) I think comparison of MAGICC with IPCC needs to be more carefully done. I am almost certain they both use the same radiative forcing expressions and hence the only reasons for differences would come from different scenarios of trace gas change, and possibly different handling of N2O overlap. (ii) wording like "MAGICC reasonably reproduces : : : AM3" (8-6) and "not captured by MAGICC" (8-11) implies to me that there is a belief that the AM3 radiation code is superior to the radiative transfer codes used to generate the RF

expressions in MAGICC and IPCC. Of course, this may be the case, but if this cannot be established with certainty, the wording should be more cautious. And as noted in my general comment (2) without knowing the size of the unforced variability in AM3 it is hard to place any differences in perspective.

**Response:** We appreciate the thoughtfulness of this comment and agree with the referee on several of their points. While the same radiation parameterizations are used for IPCC and MAGICC for carbon dioxide and methane atmospheric concentrations, we believe that it is the way that tropospheric ozone in MAGICC (and other short-lived trace gas chemistry) is simulated that is causing a difference between IPCC and MAGICC methane RFs. While minor differences exist in the definition of baseline (IPCC RFs are calculated with a 1750 forcing baseline, and MAGICC RFs is calculated with a 1765 forcing baseline but we adjust to 1860 to compare with CM3); and present-day concentrations (at 2011, $CO_2$ concentrations in IPCC and MAGICC are 391 ppm and 393 ppm, respectively; CH4 concentrations in IPCC and MAGICC are 1803 ppb and 1782 ppb, respectively), the simplified treatment of tropospheric ozone forcing in MAGICC due to virtually no spatial resolution because of hemispheric averages is very different than the vastly more sophisticated multi-model studies under Atmospheric Chemistry and Climate Model Intercomparison Project (ACCMIP) that are used in IPCC AR5. This is also one reason why we imply that the sophisticated models (such as AM3) are superior in radiation code to that by MAGICC.

We have greatly clarified and addressed all of this in the text (lines 10-24 – 11-3): "*Preindustrial to present-day forcings for $CO_2$ and methane simulated by AM3 and MAGICC are similar to those given by IPCC (Myhre et al., 2013), (1.68 from CO2 emissions and 0.97 W m-2 from methane emissions in 2011 relative to 1750 levels), albeit there are important differences, including baseline years (1750 for IPCC and 1870 for AM3 and MAGICC in this study to match that from AM3) and time series of atmospheric concentrations (Myhre et al., 2013). While the same radiation expressions are used for IPCC and MAGICC for CO2 and methane atmospheric concentrations, the representation of tropospheric ozone chemistry and its radiation effects in MAGICC is extremely simplified due to hemispheric averages in a four-box atmosphere. For a short-lived climate forcer that is highly spatially variable, this is a vastly different treatment than that by the IPCC, which employs multi-model assessments for tropospheric ozone forcings. We find that our direct methane forcing in MAGICC in model year 2011 is 0.45 W m-2, extremely close to the IPCC's forcing of 0.48 W m-2 from changes in methane concentrations alone (recall however different baselines) (Myhre et al., 2013). However, when methane interactions with other chemical species are accounted for, MAGICC estimates a forcing of 0.7 W m-2 attributed to changes in methane compared to the IPCC's value of 0.97 W m-2 (Myhre et al., 2013).*"

9-13: It is hard to know what conclusions to draw from this paragraph – is it that the

CM3 signals are unreliable because of the difficulty of extracting a small signal from a small ensemble given the size of the unforced variability (especially in the 1900-1915 period when methane forcing was less than 0.2 W m-2)?

**Response:** The purpose of this paragraph was to address two prominent, but unusual, features of temperature responses to methane. We agree with the reviewer that this was confusing and could be made a lot clearer. We have therefore clarified the text to address the features but also explain their significance for our study (lines 12-15 – 12-26): "*Two major features of the temperature response to methane in CM3, that are not present in MAGICC, further highlight the difficulty of extracting a small signal (and with a small ensemble) given the size of the unforced variability (Fig. 6); methane's forcing is considerably smaller than that of CO2, making it difficult to extract a temperature response from the variability.. The first feature is a global mean cooling response to methane forcings around 1900 to 1915, which is strongly apparent in two of the three initial condition-driven ensemble members. This cooling response is not clearly reflected in the forcings of both the direct and indirect methane responses, and while total methane forcings in AM3 are slightly negative (at most -0.15 W m-2) from 1895 to 1900, they are positive (around 0.2 W m-2 on average) from 1900 to 1915 (Figs. 3 and 4). The second feature is a strong warming signal in response to methane from 1980 to 1995, followed by cooling through 2000; while this is consistent with AM3 RFs (Fig. 3), the feature is more pronounced in the temperature response. Both of these features fall within the range of annual temperature swings due to unforced variability in CM3 (at most around 0.2 ºC for a five-year running mean). Therefore, we cannot conclude that they are robust responses to methane, but rather serve as a further example of why CCMs are difficult to employ for small individual forcings and the need for large ensembles*.*"

10-10 and especially 10-13: Again it is hard to know what conclusion to draw here, and there is nothing to lead the reader. I believe that this paragraph should come much earlier in the paper and be flagged as a major caveat when trying to extract small signals from ESMs, and in particular CM3 given its such large variability. As it is, this aspect is presented almost as an afterthought.

**Response:** We thank the referee for this suggestion, and we have moved this paragraph to the beginning of the results section, to better ground the rest of the discussion in the context of internal variability. Further, we have expanded the paragraph based on other feedback herein, and ran an additional simulation to analyze the role of unforced variability in AM3. We have also moved the last figure to the beginning of the paper, and expanded it to include AM3 as well. Throughout the text, the narrative has been modified to address

11-12: "useful threshold" – I agree with this, but I don't believe the paper applies this threshold. If it did, I think one conclusion could be that some of the early 20th century signals attributed to methane in AM3/CM3 are not robust and should not be interpreted as

such. This may be what 11-13 to 11-15 is trying to say. If so, it needs to be stated more clearly, earlier in the paper and reflected in the abstract.

> **Response:** We appreciate this feedback, and ultimately decided to remove and/or modify some of the text. It now read (lines 14-10 – 14-19): "*Further, we find that methane accounts for a considerable fraction of 20th Century and early 21st Century warming—roughly half that of CO2's warming response. However, there are some features present in CM3 results without parallels in MAGICC. The features are, however, consistent in magnitude with forcing and temperature fluctuations due to unforced variability, and therefore are unable to be classified as robust responses. A good example of this is that CM3 exhibits a cooling response to methane from 1900 to 1915 likely due to the formation of the southern ocean polynya leading to very large unforced multidecadal time-scale climate variability. This highlights how unforced variability present in sophisticated models can make it difficult to ascertain robust responses to small changes in multiple forcings individually, further justifying the use of a model such as MAGICC beyond pure accessibility. To overcome this challenge, a larger number of ensembles could be employed or simulations can be run with a quasi-chemistry-transport model (Deckert et al. 2011).*

**Comment 4:** Additional comments

Throughout: I suggest that the 4 in CH4 is subscripted throughout, for consistency with CO2.

> **Response:** We have made these changes.

Throughout: There is a lack of consistency in the labelling of simulations – compare the abbreviation column of Table 1 with Equations (1) to (4).

> **Response:** We have made the Table and Equations consistent in their labeling.

1-7: The abstract does not highlight any of the scientific results emerging from CM3 – the fact that methane-induced warming is competitive with CO2 for long periods and the high variability of its indirect effect. My other comments indicate some skepticism about these results, but it seems odd not to mention them at all in the abstract if the authors stand by them.

> **Response**: Further analysis of the unforced variability indicates that we cannot confidently say that any of these features are from the forced changes (they are all within the range of swings shown in the control runs (no external forcings)). Therefore, we have modified the text throughout the paper to make this clear, using the results (such as the fact that methane-induced warming is competitive with CO2 for long periods and the high variability of its indirect effect) as examples of why a reduced-complexity climate model is a better tool, and have

emphasized the importance of unforced variability in the abstract, especially as a justification for using MAGICC.

2-3: I suggest this sentence is re-worded to avoid the appearance of policy advocacy. The standard and effective IPCC wording is "if we want to avoid warming then we need to reduce emissions".

**Response:** We have made these changes.

2-10: I looked at Etminan's paper and could see nothing on temperature change.

**Response:** Etminan's paper substantiates the claim that methane emissions account for a quarter of today's positive radiative forcing (via radiative forcing of methane), and also provides the data for the calculation that today's anthropogenic methane emissions will have a larger impact on near-term warming than today's fossil fuel emissions (via GWP calculations based on new radiative efficiency of methane). However, we see how this was confusing and have tried to clarify what information was from each source in the text (lines X-X): "*Methane emissions in particular account for a quarter of the excess energy trapped by human emissions, and today's global anthropogenic methane emissions will have a larger impact on near-term warming than today's global fossil fuel CO2 emissions (based on forcing data provided in Myhre et al., 2013 and references therein; methane emissions provided in EPA, 2012; CO2 emissions provided in IEA, 2015; and radiative efficiency estimates of methane provided in Etminan et al., 2016)."*

2-16: For sure, the GWP concept has a lot of problems, but the strong variation of its value with time horizon (e.g. compare GWP(20) with GWP(100)) does capture "important temporal distinctions".

**Response:** This is true, however GWP is almost always employed with one time horizon selected by the user. We have clarified in the text that the way GWP is currently used (selection of a single time horizon) does not capture temporal distinctions *unless* two time horizons are employed simultaneously (e.g. Ocko et al., 2017).

4-1 and 4-7: Is it correct that MAGICC is driven by concentrations for 1765-2005 and emissions for 2006-2014? The fact that the methane lifetime is mentioned as being updated at 6-15 adds to confusion as to whether the model is concentration or emission driven. The (possible) emission/concentration confusion is also present in the paragraph after 6-15. But then 7-17 and Figure 1 seems to clearly imply that both models are concentration driven. So I ended up a bit confused.

**Response:** Yes, MAGICC is driven by concentrations up through 2005 and then emissions thereafter. We understand how that is confusing with lifetime properties being updated, and have clarified in the text, and also Figure 1.

4-15: There are significant differences in the CO2 expression used in the two references given here – I doubt that MAGICC uses the IPCC FAR expression.

**Response**: MAGICC v6 uses the IPCC FAR expression (Shine et al., 1990) but with an updated scaling parameter from Myhre et al. (1998). Please see: http://wiki.magicc.org/index.php?title=Radiative_Forcing. We have clarified this in the text.

4-25: Units of climate sensitivity missing.

**Response:** We have added the units.

5-19: A minor query, but I wasn't sure what the "indirect feedbacks of CH4 on CO2" were – it could include the fact that CH4 is oxidised to CO2, or that CH4-induced climate change impacts the carbon cycle or CH4-induced changes in O3 impact the carbon cycle by their effect on vegetation (or all three).

**Response:** We meant that CM3 CO2 concentrations do not get altered by reactions that occur in the model, and have clarified this in the text.

6-15: Coupled with my comment at 4-15, I am confused. Does MAGICC use the AR5 forcing expressions for methane or does it use the radiative efficiencies (which are W m-2 ppbv-1 normally defined for small perturbations from present day)? I don't think MAGICC can use both and this needs to be clarified.

**Response**: We agree with the referee that this is confusing, and we appreciate the careful attention to the details in our paper. The MAGICC v6 documentation (Meinshausen et al. 2011) specifies the forcing expressions when discussing radiative forcing routines for methane, and we ran additional MAGICC simulations to test the sensitivity of climate responses to changed methane radiative efficiency inputs, and they scaled proportionally. The radiative forcing expressions are used to handle the overlapping absorption bands between methane and nitrous oxide, and also take the radiative efficiency into account. We have clarified this in the text.

7-6: I found this sentence a bit cryptic. Presumably N2O and its overlap are considered in the AM3 radiation code? I was not sure why the dependence should lead to a "likely overestimate" of methane forcing.

**Response**: The radiation code does indeed account for the spectral overlap, and we have removed the confusing sentences in the paper.

8-28: I don't agree with NASA and NOAA temperature series being labelled as "independent observational datasets" when there is, of course, a huge commonality in the

underlying data, and I believe in some of the applied corrections to that data. A better wording could be used.

**Response:** We have removed the word independent.

11-1: "accurately" – the word "adequately" is used at 9-1 and I believe it a much better description of Figure 4.

**Response:** We have made this change.

11-21: One issue with MAGICC is that (as I understand it) only the executable, rather than the source code, is available, which is at odds with the title at 11-19. It may be worth making this clear to readers.

**Response:** We have clarified this in the text.

**Response:** We have made this change.

**Responses to RC2 (Anonymous Referee #2):**

**Comment 1:** Simplified models have beside the lower need of computational resources the advantage that the internal variability is small or zero and it is possible to assess the impact of small changes, while the internal variability in complex models is too large therefore. But this makes it at the same time difficult to compare them and evaluate the simplified model. As Reviewer #1 stated it is difficult to evaluate forcings which are in the same order as the unforced internal variability (Fig 7) and the variation of different ensemble members (e.g. Fig. 5, 1960). The fact that the internal variability of CM3 is very large compared to CMIP5 models, should be mentioned earlier in the text to make it easier to put the results in the right context. The text is partly formulated as CM3 is the truth and MAGICC should reproduce the same features. While this is important if the features are physically base, it is not the case if the features are due to internal variability, as the benefit of simplified models is that the results are almost free of internal variability. Additionally Fig 4 suggests that MAGICC provides better agreement with observations than CM3 does. Similar to reviewer #1 I would suggest putting more focus on the fact that it is difficult to evaluate simplified models by complex models with large variability. In addition some possible ways to overcome this problem could be provided, e.g. larger number of ensembles or simulations with a quasi-chemistry-transport model mode (e.g. Deckert et al., 2011). Publikation: Deckert, Rudolf und Jöckel, Patrick und Grewe, Volker und Gottschaldt, Klaus-Dirk und Hoor, Peter (2011) A quasi chemistry-transport model mode for EMAC. Geoscientific Model Development, 4, Seiten 195-206. Copernicus Publcations. DOI: 10.5194/gmd-4-195-2011 ISSN 1991-959X DOI: 10.5194/gmd-4-195-2011 ISSN 1991-959X

> **Response:** We thank the referee for the constructive and thoughtful feedback, and have made a lot of changes in response to these comments and related comments by Reviewer #1. Broad changes throughout the text include:
> - more emphasis on unforced variability as an additional reason why simplified models are preferred tools for impacts of small changes;
> - text modifications to caveat the difficulty in comparing simple model results with more complex models;
> - discussion of unforced variability in AM3/CM3 expanded and moved to the beginning of the results section rather than the end;
> - an additional simulation to isolate unforced variability in AM3, and the results added to the figure with the CM3 control run;
> - text modifications to reduce the impression that CM3 is "truth;" and
> - addition of strategies to overcome the challenge of unforced variability.

**Comment 2:** For my opinion the description of the models and simulations should be more detailed. I had for example some difficulties to exactly understand what the models use as an input and which parameter were calculated by the models.

> - Are the concentrations (p4-l1) or the emissions (p4-l7) prescribed in MAGICC?

**Response**: Concentrations are prescribed through 2005 in MAGICC, and then emissions drive the model. This is clarified in the text (lines 7-23 – 7-25): "*(Note that the updated atmospheric lifetime only impacts the model from 2006-2014 as it is driven by emissions and not concentrations during this period.)*" And (lines 9-2 – 9-4): "*(Note that concentrations are prescribed for MAGICC only through 2005, and then emissions inputs drive the model thereafter; however, the resulting concentrations from these emissions are consistent with that input into CM3.)*"

- Was the choice of the ensemble members of CM3 randomly or did you choose years with extreme or mean values?

    **Response:** The CM3 ensemble members were chosen stochastically. We have clarified this in the text.

- How is the RF calculated in CM3?

    **Response**: The basic shortwave and longwave radiation algorithms used in CM3 are described in Freidenreich and Ramaswamy (1999) and Schwarzkopf and Ramaswamy (1999), respectively, modified as in GAMDT (2004) to enhance computational efficiency. The shortwave algorithm includes 18 bands in the solar spectrum, and the longwave algorithm includes eight bands. Shortwave radiation parameterizations account for absorption by water vapor, carbon dioxide, ozone, molecular oxygen; molecular scattering; and absorption and scattering by aerosols and clouds. The longwave radiation parameterizations account for absorption and emission by water vapor, carbon dioxide, ozone, nitrous oxide, methane, halocarbons (CFC-11, CFC-12, CFC-13 and HCFC-22), aerosols, and clouds. Aerosols included are sulfate, carbonaceous (black and organic carbon), dust, and sea salt. Indirect effects of aerosols on clouds are included, and sulfate and black carbon are assumed to be homogenously mixed. We have added these details to the text. However, in this study, we calculate radiative forcings using the atmosphere-only component of CM3—AM3. We do this to calculate RFs that are most similar in definition to that by MAGICC.  We therefore use AM3 to diagnose transient effective radiative forcing (ERF) (the change in net radiation balance at the top-of-atmosphere (TOA) following a perturbation to the climate system taking into account any rapid adjustments (Myhre et al., 2013)) due to CO2 and methane.

    Additions to text include (lines 6-28 – 7-3): "*Shortwave and longwave radiation algorithms in CM3 are described in Freidenreich and Ramaswamy (1999) and Schwarzkopf and Ramaswamy (1999), respectively, with some modification to enhance computational efficiency (GAMDT 2004). The shortwave algorithm includes 18 bands in the solar spectrum, and the longwave algorithm includes eight bands. Shortwave*"

*radiation parameterizations account for absorption by water vapor, carbon dioxide, ozone, molecular oxygen; molecular scattering; and absorption and scattering by aerosols and clouds. The longwave radiation parameterizations account for absorption and emission by water vapor, carbon dioxide, ozone, nitrous oxide, methane, halocarbons (CFC-11, CFC-12, CFC-13 and HCFC-22), aerosols, and clouds. Aerosols included are sulfate, carbonaceous (black and organic carbon), dust, and sea salt."*

Freidenreich, S.M. and V. Ramaswamy (1999), A new multiple-band solar radiative parameterization for general circulation models, J. Geophys. Res., 104, 31, 389-31, 409.

Geophysical Fluid Dynamics Laboratory Global Atmospheric Model Development Team (GAMDT) (2004), The new GFDL global atmosphere and land model AM2-LM2: Evaluation with prescribed SST simulations, J. Clim., 17(24), 4641-4673.

Schwarzkopf, M. D. and V. Ramaswamy (1999), Radiative effects of CH4, N2O, halocarbons and the foreign-broadened H2O continuum: A GCM experiment, J. Geophys. Res., 104, 9467-9488

- Why does All Forcing in MAGICC have a large variability, while the CO2 and CH4 do not have one? Are the forcings (except CO2 and CH4) prescribed?

  **Response**: The All Forcing variability in MAGICC is due to prescribed forcings from volcanic eruptions.

- Why does CO2 show negative Forcing in Fig 2 although the concentration increases?

  **Response**: This is most likely due to unforced variability in AM3, from a mixture of land snow/ice cover variability and just year-to-year variability in the weather; soil moisture may also play a role. An additional simulation that we performed to assess the magnitude of unforced variability in AM3 revealed that Radiative forcings at the top-of-atmosphere for all-sky conditions can yield annual swings of 0.8 W m-2, and 5-yr running means dampen this to around 0.35 W m-2. Decadal swings are around 0.2 W m-2. The negative forcings despite increased concentrations falls within the realm of expected variability.

- Why are direct and indirect CH4 effects anti-correlated or have a time lag? Is there a physical explanation or is it an artifact of the internal variability?

  **Response**: We have determined that the anti-correlation and other unusual features in the figure were almost entirely due to our methodology of

double subtraction to calculate indirect methane forcings in AM3 from the CH4-only and direct CH4 forcings.

We determined this via running additional simulations to calculate the indirect methane forcings and responses via modifications to chemistry rather than our original method that introduced a double subtraction to determine the indirect effects. We ran AM3 and all ensemble members of CM3 for an experiment where methane radiation varied with time but chemistry was held at 1860 methane levels.

We have therefore updated the manuscript to reflect this new methodology for indirect effects of methane, keeping the original methodology for direct effects of methane. This method is also more consistent with how we calculated indirect effects of methane in MAGICC. While the direct and indirect effects are not completely additive (see figure below) this is typical nonlinearity effects for models of this complexity.

[Figure]

[Figure]

- Why is the temperature change of CH4 of CM3 negative although the forcing is positive?

  **Response**: Similar to CO2 forcings in the former Fig. 2, temperature responses to methane can be negative despite positive forcings due to unforced variability in CM3. We find that unforced internal dynamics (interaction among the ocean-atmosphere-land system) in CM3 introduce yearly temperature swings of 0.09 °C on average (though it can be as high as 0.4 °C), and this drops to 0.02 °C on average when five-year running means are employed (at most 0.2 °C) on average. The negative temperature responses to CH4 fall within this realm of variability. We have added text throughout the paper to discuss the role of unforced variability in influencing the results, making it difficult to compare MAGICC with CM3 and also providing more justification for why a model like MAGICC is needed.

- p6-l5 MAGICC simulates from 1750-2100, but in p4-11-9 only information about concentrations and forcings between 1765 and 2014 are given

  **Response**: We have edited the text to explain that emissions drive the model from 2005-2100 but that we restrict our analysis here through 2014.

- P6 l21 Does the 'downloaded' version of MAGICC include tuning to the multi-model mean or can be chosen which AOGCM is used for calibration?

**Response**: The downloaded version of MAGICC allows for tuning of parameters, and one can choose which AOGCM parameters to employ. We have clarified this in the text (lines 5-24 – 5-25): "*The user of the downloaded MAGICC model can select which parameters to use for each simulation.*"

- P8-25 A description about the kind of data used should be included

  **Response**: We have clarified this in the text (lines 11-22 – 11-25): "*Figure 5 shows the historical global-mean surface air temperature responses to changes in all-forcings in MAGICC and CM3 compared with NOAA and National Aeronautics and Space Administration (NASA) time series of global surface temperature anomalies data, freely available online (https://www.ncdc.noaa.gov/cag/time-series/global and https://data.giss.nasa.gov/gistemp/).*"

**Comment 3:** Technical comments

P5-l12 comma is missing after carbon dioxide

  **Response**: We have made this change.

P6-29 Here RF is defined at the tropopause, while it is defined at the top of the atmosphere in p7-23

  **Response:** We thank the referee for catching this error and we have corrected it.

P8-3 change 'slightly offset' in   Žoffset' (1W/m2 is large compared to the forcing)

  **Response**: We have made this change.

P11-1 change 'accurately' in 'adequately'

  **Response**: We have made this change.

Is there a reason why the Fig starts in different years (1860, 1870 or 1880)?

  **Response:** Yes. The CM3 climate model runs start in 1860, but the AM3 forcing runs start in 1870 due to constraints by the prescribed SST observations. Observational data for global surface air temperature begins in 1880.

**Responses to SC1 (Marcus Sarofim):**

**Comment 1:** MAGICC is, of course, one of the most widely used tools for this purpose, and therefore a reasonable choice. However, it might be worthwhile for the authors to discuss a couple of alternatives.

a. Hector is in a similar class of model as MAGICC, but has the advantage of being fully open-source (see comment on 11-21 from Referee 1). Hector is described in Hartin et al. 2014, https://core.ac.uk/download/pdf/25503085.pdf. Also relevant is a thesis by Schwarber on comparing Hector and MAGICC at https://www.atmos.umd.edu/theses_archive/2016/aschwarber_masters.pdf. I am not suggesting running Hector for this paper, as that would be a large lift, but a citation and brief mention of the benefits of open source could be worthwhile.

b. Meanwhile, there are approaches that are even simpler than MAGICC. Melvin et al. (2016), for example, estimated the physical impacts resulting from methane mitigation by using the simplified expressions from AR5 for concentration and radiative forcing and from Shine et al. (2005) for temperature. It could be a valuable sensitivity analysis to take one of these simplified approaches as an additional comparison. These simplified equations may be better suited to analyzing the marginal effect of perturbations in emissions, rather than to simulate overall global temperature change from total anthropogenic emissions: however, it seems to me that this kind of marginal analysis is consistent with the goals of this paper.

c. The NAS in their report on valuing climate damages (https://www.nap.edu/catalog/24651/valuing-climate-damages-updating-estimationof-the-social-cost-of) suggest the use of the FAIR model: this would fall between Hector/MAGICC and the GTP-style equations in terms of complexity.

> **Response:** We thank the reviewer for the thoughtful comments and additional information. We have added text to acknowledge the other tools that exist with complexity levels between GWP and GCMs, with the above references, and have made it clear that MAGICC is not the only tool in its class. However, we do note that none of the above options seem appropriate for our particular analysis: Hector does not explicitly calculate concentrations of methane, and uses input files instead; GTP expressions, which we have used previously and compared with MAGICC, do not allow for a changing atmospheric lifetime in response to changing OH concentrations – a critical feedback for methane impacts; and finally, the FAIR model goes beyond the climate indicators we are interested in, by including a cost sub-model to look at economic parameters. Therefore, we did not include specific details about these tools in the text.

**Comment 2:** As with Referee 1, I find it surprising that the AM3 forcing results only diverge in the last couple decades in contrast with MAGICC which shows a slowly growing divergence over the entire run. Would there be any effect of running AM3 with different initial conditions, which could show whether this is a robust result or a result

deriving from internal variability? The only other explanation that comes to mind other than initial condition sensitivity is that somehow CO2 and CH4 forcing have differential sensitivity to SSTs or sea-ice extent. Maybe a constant-concentration experiment could be informative in terms of whether the forcing of methane and CO2 might respond differently to the historical changes of SST & sea-ice?

> **Response**: We thank the reviewer for these thoughts. We ran an additional simulation with constant preindustrial (1860) external radiative forcings and repeating seasonal cycle of sea surface temperatures and sea ice characteristics for 200 years for AM3 in order to assess the role of unforced variability. Analysis revealed that radiative forcings at the top-of-atmosphere for all-sky conditions can yield annual swings of 0.8 W m-2, and 5-yr running means dampen this to around 0.35 W m-2. Decadal swings are around 0.2 W m-2. Therefore, the CO2 and methane forcing swings, especially at smaller forcings, fall within the realm of variability and therefore cannot be considered a robust feature. We have added this discussion to the text.

**Comment 3:** I do find this comparison of more complex models to simple models to be an informative exercise (see Sarofim 2012 where I used both the MIT IGSM and MAGICC to calculate the 100-year sustained GTP for methane: https://link.springer.com/content/pdf/10.1007%2Fs10666-011-9287-x.pdf). But, as Referee 1 notes, this comparison is complicated by the variability inherent in more complex models, even as at the same time, this is one of the motivators behind the use of simple models for investigating the effect of emissions perturbations that are expected to have temperature effects smaller than the internal variability of the complex models. I don't have a good answer for this, other than averaging even larger ensembles in order to reduce initial-condition-based noise even more.

> **Response:** We wholeheartedly agree with the reviewer. We have substantially expanded our discussion and analysis of the role of unforced variability in not just this analysis, but in comparison between simple and more complex models in general. We also addressed more directly the fact that this is indeed a major motivation for using simpler models for smaller forcing changes. With present computational resources, more ensemble members to diagnose multiple individual forcings and resulting temperature responses is not feasible. However, despite temperature swings in CM3 unparalleled by MAGICC, the general trends and magnitudes are fairly consistent, providing confidence in our use of MAGICC going forward.

[revised manuscript text omitted]

---

## Referee Report (RR1)

Comments on revised version of "Rapid and reliable assessment of methane impacts on climate" by I. Ocko et al.

**Prologue**

I have entered the review process only at its final stage, replacing an earlier referee. Hence, my original intention had been just to check, whether the responses of the authors to the referee comments from the main stage have been adequate, and whether the paper text has reached a stage of maturity. This intended limitation of my referee status notwithstanding, I perceive shortcomings of this paper to an extent that I am unable to bring myself to passing over in silence.

**Recommendation**

My recommendation is to return the manuscript to the authors once again, at least to do some more 'polishing' on the presentation. I also strongly support to make a title change along the line of anonymous referee1's suggestion: "The difficulty of using small ensembles of simulations of an ESM with large interannual variability to validate simple climate models in cases of small forcings"

**Reasons**

I regard the main conclusions of the paper, 1) 'Well trained simplified models like MAGICC are able to provide a representation of the global mean climate well enough to provide assessment studies', and 2) 'Complex climate models have limited ability to identify the response to small forcings in cases where the expected response and the simulated internal variability have similar orders of magnitude', as basically correct. However, the authors have allowed repeatedly to let themselves get carried away by their enthusiasm, and have inflicted into their paper a number of exaggerations of alleged complex model disadvantages that ought to be toned down for the final manuscript. (Beyond this, some confusing formulations need to be rectified.)

I also feel that the ESM simulations have not been optimally setup for the fairest possible comparison with MAGICC. Besides the possibilities, which specified dynamics simulations (e.g., Lamarque et al., 2012; Kooperman et al., 2012) would have been offered for reducing ESM internal variability, the possibility to calculate radiative forcing (RF) rather than effective radiative forcing (ERF) by radiation double calling (e.g., Chung and Soden, 2015; Dietmüller et al., 2016) has also not been exploited. This implies that when discussing radiative forcing calculations, the authors' comparison is often not so much between a noisy ESM and a noise-free simplified model, but rather between a noisy ERF and a noise-free RF (see extensive discussion in Forster et al., 2016). However, as the main referees have not been so strict, I won't go nitpicking either, here. Yet, I request that the distinction between RF and ERF ought to be clear throughout the paper, and that the consequences of using ERFs from the ESM, but RFs from MAGICC are openly discussed.

Specific Remarks

p. 1, l. 13 (Abstract): I recommend to phrase more carefully as follows: 'Using basic knowledge from observations and complex Earth system models, reduced-complexity climate models offer an ideal compromise in that they provide quick reliable insights into climate responses, with only a limited computational infrastructure needed. They are particularly useful for simulating the response to forcings of small changes in different climate pollutants, due to the absence of mentionable internal variability.'

p. 2, l. 23: Please, cite Fuglestvedt et al. (2010) in addition.

p. 4 l. 6, 7: '… if … the response is comparable..' ; '… and (iii) whether the lack of internal variability …'

p. 6, l. 17: 'climate sensitivity of MAGICC'; confusing as it is variable and arbitrarily set. Either recall the mean value from p. 5, l.22, or state as the reason is that CM3 has a higher sensitivity than the majority of AOGCMs used to define the MAGICC climate sensitivity).

p. 8 , l. 16 (major issue): ERF is introduced here as an extra abbreviation but is not used later when the AM3 forcings (p. 10, l. 7) are presented which are confusingly designated with RF. Please, rectify throughout the paper.
l. 18: delete one bracket behind Myhre et al. and, please, cite Shine et al. (2003) in addition.

p. 8, l. 30 Please, notice a technical error with the equation setting.

p. 10, l. 7: (major issue): Following Forster et al. (2016), ERF can hardly be expected to take robust values when calculated on a one-year or 5-year basis. Thus, the MAGICC vs. AM3 comparison must remain on a plausibility level within this paper, which should be emphasized.

p. 10 l. 25: Change CO2 to $CO_2$ (several further examples afterwards).

p.10, l. 20: I guess it's AM3 rather than CM3?

p. 11, l. 1-3: Confusing, as you have given another number (0.97 W/m$^2$) for the methane forcing on p. 10, l. 25. Is this *exclusively* due to the different reference year? If yes, please state so clearly; if no, please give other potential origin(s) for the issue.

p. 11, l. 13 : In my understanding, here we are not dealing with 'responses', but with 'forcings' and 'adjustments' from concentration changes; the 'response' is coming only in the next subsection, so please adjust the phrasing.

p. 11, l. 20: Throughout this subsection, the issue of differing climate sensitivity between CM3 and the MAGICC mean is not raised, hence creating the impression that you would regard identical surface temperature responses between the models as an optimal evaluation result. This is obviously not the case, please reconsider.

p.11, l. 29 (major issue): "MAGICC has much higher correlation coefficients [with observed data], likely through the absence of internal variability". This I an odd sentence

which urgently needs to be set in an adequate context. Of course, reality has no "internal variability" because there is but one realization of it! Hence, it is absolutely necessary that such an argument must not even hint at the fact that MAGICC provides better agreement with reality than CM3 (or any other complex model) does.

p. 12, l. 6: 'correlation of the ensembles means', recall that the MAGICC ensemble is for 19 different model representations, while CM3 has 3 independent realizations of the same model.

p. 12, l. 19 (major issue): Given that the temperature response lags the radiative forcing, I deem it not surprising that (spurious) negative radiative forcing, while leading to temperature decrease after 1895, may not correlate with negative temperature response during the same time period, but occurs only somewhat delayed.

p. 12, l. 27 until end of this section: As the authors (correctly) claim a far-reaching influence of internal variability on the exact evolution of the temperature response time series simulated by CM3, I see no much sense in looking for mechanistic reasons to explain details of the actual evolution. If you like to keep this, please motivate it more convincingly. Another example of negative temperature responses simulated in case of $CO_2$ increase has been given by Huszar et al. (2013, their Fig. 10). There, too, mechanistic explanation attempts would have been obsolete.

p. 13, l 21 (major issue): "sophisticated coupled chemistry-climate models … are generally unsuitable for analysis of methane mitigation strategies"; expressed with such dogmatic universality, this statement has to be rejected. It only holds, if assessments of many mitigation options – especially with relatively little expected impact -, or a large extent of parameter sensitivities are to be investigated. Then application of complex models becomes clearly unfeasible and resorting to simplified models is doubtlessly required; best may be a combination of both as, e.g., described by Dahlmann et al. (2016) for an example of aviation impact mitigation.

References:

Chung, E.-S., Soden, B.J., 2015: An assessment of methods for computing radiative forcing in climate models, Environ. Res. Lett, 10., 074004.

Dahlmann, K., et al., 2016: Can we reliably assess climate mitigation options for air traffic scenarios despite large uncertainties in atmospheric processes? Transport. Res. Part D, 46, 40-55.

Dietmüller, S., et al., 2016: A new radiation infrastructure for the Modular Earth Sub-model System (MESSy, based on version 2.51), Geosci. Model Dev., 9, 2209-2222.

Forster, P.M., et al., 2016: Recommendations for diagnosing effective radiative forcing from climate models for CMIP6, J. Geophys. Res. 121, 12460-12475.

Fuglestvedt, J., et al., 2010: Transport impacts on atmosphere and climate: Metrics, Atmos. Environ., 33, 4648-4677.

Kooperman, G.J., et al., 2012: Constraining the influence of natural variability to improve estimates of global aerosol indirect effects in a nudged version of the Community Atmosphere Model, 5, J. Geophys. Res. 117, D23204.

Lamarque, J.-F., et al. 2012: CAM-chem: description and evaluation of interactive atmospheric chemistry in the Community Earth System Model, Geosci. Model Dev., 5, 369-411.

Huszar, et al., 2013: Modeling the present and future impact of aviation on climate: an AOGCM approach with online coupled chemistry, Atmos. Chem. Phys. 13, 10027-10048.

Shine, K.P., et al., 2003: An alternative to radiative forcing for estimating the relative importance of climate change mechanisms, Geophys. Res. Lett., 30, 2047.

---

## Author Response (AR2)

**Manuscript Ref: acp-2018-26**

**Rapid and reliable assessment of methane impacts on climate**

Ilissa B. Ocko, Vaishali Naik, and David Paynter

We sincerely appreciate the time the referees have taken to carefully review the manuscript revisions in response to the initial comments. The additional suggestions have certainly strengthened the paper. Below, we respond point-by-point to comments (reviewer comments in blue, responses in black).

**Responses to Anonymous Referee #1:**

**Comment 1:** I have been rather sensitised by a comment from another reviewer to the fact that this paper should not be seen to specifically endorse MAGICC (as it can be read to be doing so in several places) but more generally endorse the use of reduced complexity models, of which MAGICC is just one. So for example, at 1:21 it would be better written as "reduced complexity models such as MAGICC", and similarly at 4:8-9, 13:18-19 and 14:20.

> **Response:** We appreciate the feedback and agree with the reviewers. We have tweaked the language in all of the locations identified.

**Comment 2:** In their response to my original comment 11:21, the authors said they would note that MAGICC is not open source. They say they have acted on this, but I couldnt find where they had done so (unless it is the cryptic (to some) statement that the executable is available (14:26)). For some people, this is a major limitation of using MAGICC because, frankly, one doesnt fully know what is really in it. I suggest that in the "limitations" around 5:27 and again at 14:25 this issue is explicitly noted. It may help to encourage the custodians of MAGICC to enter the 21st century!

> **Response:** We wholeheartedly agree that the closed source code is a big limitation and frustration to using MAGICC, and have more clearly stated that it is not open source beyond the available executable statement. We have added statements in both locations identified by the reviewer (both in the methods and conclusions sections). The reason why we use MAGICC is because of its already widespread use in the community for mitigation analyses (such as Shoemaker et al., 2013 and World Energy Outlook 2017).
>
> Shoemaker, J. K., Schrag, D. P., Molina, M. J. and Ramanathan, V.: What role for short-lived climate pollutants in mitigation policy?, Science, 342(6164), 1323–1324, 2013.

**Comment 3:** At 6:10 the authors say that the ECS is higher than MAGICC, but as I understand, it is in the range of the calibration models - so strictly it is higher than the MAGICC median and mean? This could be usefully clarified.

> **Response:** This is an excellent point and we have clarified. The text now reads (lines 6:12-13 in tracked changes version): *"The equilibrium climate sensitivity of CM3 is 4.8 K (Paynter et al., 2018), which is in the range of the MAGICC calibration models but higher than the median and mean."*

**Comment 4:** Regarding my original 6-15 (I dont think this really needs acting on, as it is arguably "geeks corner") I still do not understand why MAGICC needs methane radiative efficiencies. It uses the methane radiative forcing expression, and these efficiencies are implicit in that expression. It could be that it only uses them to compute the impact of ozone change on methane.

**Response:** The MAGICC model allows the user to specify the coefficient in the radiative forcing expression, which, although is in units of Wm-2/ppb CH4, we had incorrectly referred to it as the radiative efficiency. The coefficient remains unchanged from IPCC AR4 to AR5 and therefore we did not update this parameter. We have clarified this in the text.

**Comment 5:** My original comment 8-28 suggested removing "independent" as these temperature datasets are certainly not independent. The authors seemed to concur in their response, but "independent" is still there in the text.

**Response:** We removed the word independent in the original manuscript 8-28 ("The two independent observational datasets are perfectly correlated (r = 1.00)." reads "The two observational datasets are perfectly correlated (r = 1.00)." in the submitted revised version.) However, the word "independent still exists in the previous paragraph, which we believe is what the referee is indicating. We have now removed this "independent" as well.

**Comment 6:** Minor - can be picked up at proof stage

3:17 "Raper"
5:21 "homogenEous"
8:21 The equations are a bit jumbled (two on the same line) at least in my printed version.
7:17 missing closing parenthesis
8:25 and 29 Some subscripting and superscripting needed
A few of the references are concatenated with theprevious one (Collins et al. 2016, Delworth et al. 2016)

**Response:** We thank the referee for the careful reviews and have revised the manuscript accordingly.

**Comment 1:** My recommendation is to return the manuscript to the authors once again, at least to do some more 'polishing' on the presentation. I also strongly support to make a title change along the line of anonymous referee1's suggestion: "The difficulty of using small ensembles of simulations of an ESM with large interannual variability to validate simple climate models in cases of small forcings"

> **Response:** We appreciate the careful review of our manuscript and the helpful suggestions. We have considered several titles and title changes, but ultimately prefer a simpler and more general title to a longer and more specific one. Although the referee's suggestion is ultimately one of our results, it is not the main motivation of the paper and does not capture the full scope of our study, which is to identify a quick and reliable method for methane mitigation analysis.

**Comment 2:** I regard the main conclusions of the paper, 1) 'Well trained simplified models like MAGICC are able to provide a representation of the global mean climate well enough to provide assessment studies', and 2) 'Complex climate models have limited ability to identify the response to small forcings in cases where the expected response and the simulated internal variability have similar orders of magnitude', as basically correct. However, the authors have allowed repeatedly to let themselves get carried away by their enthusiasm, and have inflicted into their paper a number of exaggerations of alleged complex model disadvantages that ought to be toned down for the final manuscript. (Beyond this, some confusing formulations need to be rectified.)

> **Response:** We have attempted to tone down the language involving disadvantages of complex models. See responses to Comment 4 for specific changes in the abstract, results, and conclusion text relevant to this comment.

**Comment 3:** I also feel that the ESM simulations have not been optimally setup for the fairest possible comparison with MAGICC. Besides the possibilities, which specified dynamics simulations (e.g., Lamarque et al., 2012; Kooperman et al., 2012) would have been offered for reducing ESM internal variability, the possibility to calculate radiative forcing (RF) rather than effective radiative forcing (ERF) by radiation double calling (e.g., Chung and Soden, 2015; Dietmüller et al., 2016) has also not been exploited. This implies that when discussing radiative forcing calculations, the authors' comparison is often not so much between a noisy ESM and a noise-free simplified model, but rather between a noisy ERF and a noise-free RF (see extensive discussion in Forster et al., 2016). However, as the main referees have not been so strict, I won't go nitpicking either, here. Yet, I request that the distinction between RF and ERF ought to be clear throughout the paper, and that the consequences of using ERFs from the ESM, but RFs from MAGICC are openly discussed.

> **Response:** We calculated ERF for a few reasons. First, it is the preferred measure of forcing since AR5 and the standard way in which forcing will be estimated for CMIP6/AR6. Therefore, we do not have radiation double calling implemented in

AM3. Second, a limitation of the double call is that it does not account for any fast adjustments to the climate system that occur due to the forcing agent (i.e. stratospheric cooling, fast cloud responses). These can impact the value of forcing considerably. So while using the double call would have resulted in much less noise in the forcing signal for sure, it would have also been an estimate of the forcing which is not in keeping with how forcing is generally estimated from GCM experiments or the CMIP6 protocol.

We have gone through the manuscript to clarify the distinction between AM3 ERF and MAGICC RF throughout, and have added text about the implications of using ERF in AM3 and RFs in MAGICC (lines 8:24-27 in tracked changes version): *"While RF does not capture the full alterations in the energy balance, ERF is more uncertain than RF because it involves multiple climate interactions (Forster et al., 2016). However, several studies have found that ERF and RF are nearly equal for many situations, and especially for increased concentrations in CO2 and methane (Myhre et al., 2013)."*

**Comment 4:** Specific Remarks

p. 1, l. 13 (Abstract): I recommend to phrase more carefully as follows: 'Using basic knowledge from observations and complex Earth system models, reduced-complexity climate models offer an ideal compromise in that they provide quick reliable insights into climate responses, with only a limited computational infrastructure needed. They are particularly useful for simulating the response to forcings of small changes in different climate pollutants, due to the absence of mentionable internal variability.'

> **Response:** We have modified this sentence in the abstract based on the suggestion. The text now reads (lines 1:13-16 in tracked changes version): *"On the other hand, reduced-complexity climate models that use basic knowledge from observations and complex Earth system models offer an ideal compromise in that they provide quick, reliable insights into climate responses, with only limited computational infrastructure needed. They are particularly useful for simulating the response to forcings of small changes in different climate pollutants, due to the absence of internal variability."*

p. 2, l. 23: Please, cite Fuglestvedt et al. (2010) in addition.

> **Response:** We have added this reference.

p. 4 l. 6, 7: '… if … the response is comparable..' ; '… and (iii) whether the lack of internal variability …'

> **Response:** We have made this revision.

p. 6, l. 17: 'climate sensitivity of MAGICC'; confusing as it is variable and arbitrarily set. Either recall the mean value from p. 5, l.22, or state as the reason is that CM3

has a higher sensitivity than the majority of AOGCMs used to define the MAGICC climate sensitivity).

> **Response**: The text now reads (lines 6:12-13 in tracked changes version): *"The equilibrium climate sensitivity of CM3 is 4.8 K (Paynter et al., 2018), which is in the range of the MAGICC calibration models but higher than the median and mean."*

p. 8 , l. 16 (major issue): ERF is introduced here as an extra abbreviation but is not used later when the AM3 forcings (p. 10, l. 7) are presented which are confusingly designated with RF. Please, rectify throughout the paper.

> **Response**: We thank the reviewer for this observation, and have gone through the text to make it clear that AM3 forcings when presented are ERF.

l. 18: delete one bracket behind Myhre et al. and, please, cite Shine et al. (2003) in addition.

> **Response:** We have added this reference and made this revision.

p. 8, l. 30 Please, notice a technical error with the equation setting.

> **Response:** We have made this revision.

p. 10, l. 7: (major issue): Following Forster et al. (2016), ERF can hardly be expected to take robust values when calculated on a one-year or 5-year basis. Thus, the MAGICC vs. AM3 comparison must remain on a plausibility level within this paper, which should be emphasized.

> **Response**: We have revised the text to read (lines 10:19-21 in tracked changes version): *"While these are the standard forcing calculation methods for both types of models, we emphasize that comparing values of AM3 ERF to MAGICC RF can only allow for comparisons in broad patterns and relative magnitudes, especially because of the large variability in ERF values when averaged over one to five year timescales (Forster et al., 2016), as discussed above."*

p. 10 l. 25: Change CO2 to CO2 (several further examples afterwards).

> **Response:** We have made these revisions.

p.10, l. 20: I guess it's AM3 rather than CM3?

> **Response:** It is supposed to be AM3. We have made this revision.

p. 11, l. 1-3: Confusing, as you have given another number (0.97 W/m2) for the methane forcing on p. 10, l. 25. Is this exclusively due to the different reference year? If

yes, please state so clearly; if no, please give other potential origin(s) for the issue.

> **Response:** The value cited on 10-25 is the IPCC value for RF in response to methane emissions, which accounts for both direct and indirect effects. This is the same value that is provided on 11-3. We have clarified that the value on 10-25 refers to IPCC value and not MAGICC's, as we believe this may have been the source of the confusion.

p. 11, l. 13 : In my understanding, here we are not dealing with 'responses', but with 'forcings' and 'adjustments' from concentration changes; the 'response' is coming only in the next subsection, so please adjust the phrasing.

> **Response**: We have modified the text to read (lines 11:26-27 in tracked changes version): *"The results from AM3 further highlight the role of unforced variability in complicating perceived forcings from small concentration changes."*

p. 11, l. 20: Throughout this subsection, the issue of differing climate sensitivity between CM3 and the MAGICC mean is not raised, hence creating the impression that you would regard identical surface temperature responses between the models as an optimal evaluation result. This is obviously not the case, please reconsider.

> **Response**: We have modified the text to read (lines 12:26-28 in tracked changes version): *"Further, recall that the equilibrium climate sensitivity in CM3 is larger than the mean/median in MAGICC, and therefore we expect differences in the ensemble member-averaged responses from this characteristic alone."*

p.11, l. 29 (major issue): "MAGICC has much higher correlation coefficients [with observed data], likely through the absence of internal variability". This I an odd sentence which urgently needs to be set in an adequate context. Of course, reality has no "internal variability" because there is but one realization of it! Hence, it is absolutely necessary that such an argument must not even hint at the fact that MAGICC provides better agreement with reality than CM3 (or any other complex model) does.

> **Response**: In this sentence, we meant that the lack of internal variability in MAGICC, and thus a smoother response, is likely why the correlation between MAGICC and the observational datasets is higher. We have rephrased the text as to just state objectively that MAGICC r is higher than CM3 r (lines 12:11-13 in tracked changes version): *"MAGICC and CM3 both have high correlations with NOAA and NASA data, although MAGICC's are higher (MAGICC r = 0.92 (NOAA) and 0.93 (NASA); CM3 r = 0.76 (NOAA) and 0.75 (NASA))."*

p. 12, l. 6: 'correlation of the ensembles means', recall that the MAGICC ensemble is for 19 different model representations, while CM3 has 3 independent realizations of the same model.

**Response**: We have qualified the text via the following (lines 12:22-23 in tracked changes version)*: "The correlations of the ensemble-means (19 physics-driven ensemble members for MAGICC and three initial condition-driven ensemble members for CM3) are extremely high (CO2 r = 0.98; methane r = 0.92)."*

p. 12, l. 19 (major issue): Given that the temperature response lags the radiative forcing, I deem it not surprising that (spurious) negative radiative forcing, while leading to temperature decrease after 1895, may not correlate with negative temperature response during the same time period, but occurs only somewhat delayed.

**Response**: This is a good point by the referee, and we have modified the text accordingly (lines 13:7-10 in tracked changes version): *"This cooling is likely a lagged response to negative methane ERF (at most -0.15 W m$^{-2}$) from 1895 to 1900, seen both in the direct and indirect methane forcings (Figs. 3 and 4)."*

p. 12, l. 27 until end of this section: As the authors (correctly) claim a far-reaching influence of internal variability on the exact evolution of the temperature response time series simulated by CM3, I see no much sense in looking for mechanistic reasons to explain details of the actual evolution. If you like to keep this, please motivate it more convincingly. Another example of negative temperature responses simulated in case of CO2 increase has been given by Huszar et al. (2013, their Fig. 10). There, too, mechanistic explanation attempts would have been obsolete.

**Response**: We have removed this explanation from the text.

p. 13, l 21 (major issue): "sophisticated coupled chemistry-climate models … are generally unsuitable for analysis of methane mitigation strategies"; expressed with such dogmatic universality, this statement has to be rejected. It only holds, if assessments of many mitigation options – especially with relatively little expected impact -, or a large extent of parameter sensitivities are to be investigated. Then application of complex models becomes clearly unfeasible and resorting to simplified models is doubtlessly required; best may be a combination of both as, e.g., described by Dahlmann et al. (2016) for an example of aviation impact mitigation.

**Response**: We understand where the referee is coming from, and have rephrased the text (lines 14:12-16 in tracked changes version)*: "
[revised manuscript text omitted]
) r̶a̶d̶i̶a̶t̶i̶v̶e̶ forcings (W m⁻²) a̶f̶t̶e̶r̶ ̶s̶t̶r̶a̶t̶o̶s̶p̶h̶e̶r̶i̶c̶ ̶a̶d̶j̶u̶s̶t̶m̶e̶n̶t̶,̶ ̶f̶o̶r̶ ̶b̶o̶t̶h̶ ̶A̶M̶3̶ ̶(̶d̶a̶s̶h̶e̶d̶)̶ ̶a̶n̶d̶ ̶M̶A̶G̶I̶C̶C̶ ̶(̶s̶o̶l̶i̶d̶)̶ ̶m̶o̶d̶e̶l̶ ̶s̶i̶m̶u̶l̶a̶t̶i̶o̶n̶s̶.̶ ̶A̶M̶3̶ r̶a̶d̶i̶a̶t̶i̶v̶e̶ ̶f̶o̶r̶c̶i̶n̶g̶s̶ ̶a̶r̶e̶ ̶t̶e̶c̶h̶n̶i̶c̶a̶l̶l̶y̶ ̶'̶e̶f̶f̶e̶c̶t̶i̶v̶e̶'̶ ̶r̶a̶d̶i̶a̶t̶i̶v̶e̶ ̶f̶o̶r̶c̶i̶n̶g̶s̶,̶ ̶a̶n̶d̶ ̶i̶n̶c̶l̶u̶d̶e̶ ̶t̶r̶o̶p̶o̶s̶p̶h̶e̶r̶i̶c̶ ̶a̶d̶j̶u̶s̶t̶m̶e̶n̶t̶s̶ ̶a̶s̶ ̶w̶e̶l̶l̶,̶ ̶a̶n̶d̶ ̶a̶r̶e̶ ̶c̶a̶l̶c̶u̶l̶a̶t̶e̶d̶ ̶a̶t̶ t̶h̶e̶ ̶t̶o̶p̶-̶o̶f̶-̶a̶t̶m̶o̶s̶p̶h̶e̶r̶e̶.̶ ̶M̶A̶G̶I̶C̶C̶ ̶r̶a̶d̶i̶a̶t̶i̶v̶e̶ ̶f̶o̶r̶c̶i̶n̶g̶s̶ ̶a̶r̶e̶ ̶c̶a̶l̶c̶u̶l̶a̶t̶e̶d̶ ̶a̶t̶ ̶t̶h̶e̶ ̶t̶r̶o̶p̶o̶p̶a̶u̶s̶e̶.̶ ̶A̶M̶3̶ ̶d̶a̶t̶a̶ ̶a̶r̶e̶ ̶5̶-̶y̶e̶a̶r̶ ̶r̶u̶n̶n̶i̶n̶g̶ ̶m̶e̶a̶n̶s̶.̶ C̶o̶r̶r̶e̶l̶a̶t̶i̶o̶n̶ ̶c̶o̶e̶f̶f̶i̶c̶i̶e̶n̶t̶ ̶b̶e̶t̶w̶e̶e̶n̶ ̶M̶A̶G̶I̶C̶C̶ ̶a̶n̶d̶ ̶A̶M̶3̶ ̶f̶o̶r̶c̶i̶n̶g̶s̶ ̶a̶r̶e̶ ̶a̶l̶s̶o̶ ̶s̶h̶o̶w̶n̶.̶

[Figure]

**Figure 5. All forcing global-mean surface air temperature responses in ºC for CM3 (solid red line) and MAGICC (solid grey line) model simulations as compared to observations by NOAA (+) (https://data.giss.nasa.gov/gistemp/) and NASA (x) (https://www.ncdc.noaa.gov/cag/time-series/global). All annual temperature anomalies shown as change from 20th Century average for each dataset. Individual initial condition-driven ensemble members for CM3 runs shown in thin dashed red lines. Physics-driven ensemble-member range for MAGICC shown as shaded grey. CM3 data are 5-year running means.**

[Figure]

**Figure 6. Global mean surface air temperature responses in ºC for CM3 (dashed line) and MAGICC (solid line) model derived simulations – $CO_2$-only (orange) and methane-only (blue). Individual initial condition-driven ensemble members for CM3 runs shown in thin dashed lines. Range for MAGICC physics-driven ensemble members shown in shaded colours. CM3 data are 5-year running means. Correlation coefficient between MAGICC and CM3 temperature responses are also shown.**

[Figure]

**Figure 7. Regional surface air temperature responses in ºC for CM3 (dashed line) and MAGICC (solid line) model indirect simulations – CO2-only (orange) and methane-only (blue). Individual initial condition-driven ensemble members for CM3 runs shown in thin dashed lines. Range for MAGICC physics-driven ensemble members shown in shaded colours. CM3 data are 5-year running means. Correlation coefficient between MAGICC and CM3 temperature responses are also shown.**